# ADAPTIVE STIMULATION & RESPONSE MODELING OF LATENT NEURAL DYNAMICS

## ABSTRACT

Latent neural dynamics are a widely used model in neuroscience for describing the time evolution of collective neural activity. These models have been established as useful for neural decoding: for example, latent dynamical models of neural activity give state-of-the art predictions of ongoing kinematics in motor tasks. Despite their utility, the causal mechanisms behind the effectiveness of latent variable models remain poorly understood. To uncover how such latent variables causally encode behaviors, or how they change, would require methods for stimulating neural dynamics during an experiment. Algorithms to *drive* neural dynamics remain limited, however, due to the need to continually track and respond to changes in neural activity, to account for variation in neural responses under stimulation, and to select useful stimulations to apply from an extensive set of possibilities. Here, we develop a novel streaming method for stimulation-response modeling in affine latent spaces and an optimization framework for selecting high-dimensional stimulation patterns to drive low-dimensional dynamics. Our method integrates streaming latent space construction, an adaptive nonparametric model of the effects of stimulations, and projection maximization under feasibility constraints to determine stimuli that move dynamics along a desired vector. We demonstrate our approach on both simulated and real neural data (calcium fluorescence images, intracortical electrophysiological recordings). We evaluate our method across multiple latent space representations and multiple models of dynamics in parallel, and additionally provide a novel streaming estimator to determine which representation is most predictive of ongoing neural dynamics at any timepoint. This allows for direct comparison between different latent representations and the opportunity for adaptive selection of stimulations to best distinguish amongst neural subspace hypotheses. Finally, we demonstrate algorithm runtimes at faster than real-time speeds ($<100$ ms), making it compatible with future *in vivo* applications.

## 1 INTRODUCTION

Models of neural activity in low-dimensional spaces, often called 'neural manifolds', are increasingly state-of-the-art for describing the structure of the neurological activity that gives rise to ongoing behavior (Saxena & Cunningham, 2019; Vyas et al., 2020). Such neural population models have been very successful across areas in neuroscience, from determining latent task variables in decision-making (Peixoto et al., 2021) to decoding latent neural activity for predicting desired movements in brain-machine interfaces (Pandarinath et al., 2018). Additional developments in targeted stimulation technology have opened the door to causally testing underlying manifold hypotheses by manipulating the activity of individual and sets of neurons (Grosenick et al., 2015; Rajasethupathy et al., 2016; Jazayeri & Afraz, 2017; Tafazoli et al., 2020; Dal Maschio et al., 2017; Vinograd et al., 2024). For example, neuroscientists could test whether a pattern of neural sates forms a ring attractor via stimulating along or off the manifold in a targeted way. (Kim et al., 2017). Such higher-resolution stimulation technology is also being developed for clinical applications, where driving activity in a brain circuit has therapeutic benefits (Yang et al., 2021; Shah et al., 2024). As the number of possible stimulation targets or parameters grows, however, it becomes more challenging to determine ideal or even useful stimulation patterns. Selecting even just 30 neurons to stimulate from a population of 400 involves searching a space of over $10^{45}$ combinations, without considering stimulus magnitudes. Designing stimulations to manipulate latent neural dynamics additionally requires considering the

time-dynamics of the system: a stimulation applied early in a trial and the same stimulation applied late in a trial could have vastly different effects due to an evolution in the underlying neural state. We therefore need a method for tracking activity in latent spaces, modeling the response to potentially high-dimensional stimulations at different locations on the manifold, and finally designing a stimulation customized to in-the-moment neural dynamics.

**Prior work** has addressed specific elements of the problem of tracking neural dynamics and designing neural stimulations (Peixoto et al., 2021; Minai et al., 2024; SoldadoMagraner et al., 2025; Wagenmaker et al., 2024; O'Shea et al., 2022; Draelos & Pearson, 2020; Draelos et al., 2021). Designing stimulations from a high-dimensional set of possibilities is a significant challenge, and has been partially addressed using methods like input-output dynamical modeling (Yang et al., 2021) or Bayesian optimization (Minai et al., 2024). In many cases, spatial correlations, as in a 2D array for electrical microstimulation, can serve to reduce the complexity of the problem. In contrast, here we specifically target the situation where many neurons are at least somewhat individually addressable, as in the case of holographic optogenetic photostimulation (Adesnik & Abdeladim, 2021; Pégard et al., 2017; Triplett et al., 2023). Actively learning from the results of stimulations can also be used to design better or more custom future stimulations, as demonstrated with techniques like active learning (Wagenmaker et al., 2024), or Bayesian variational inference (Draelos & Pearson, 2020).

**Our core contribution** is a novel real-time method for designing neural stimulations that perturb latent dynamics in arbitrary directions. We propose a new model for learning a map between stimulations and their effects on latent neural dynamics. Using kernel regression, we nonparametrically regress changes in dynamics based on both the delivered stimulation and the neural latent state (location on the manifold) at the time of stimulation. We do not assume that responses to stimulations are robust, involve the neurons that the stimulation intended to target, or are static over time. We consider multiple possible models of these latent neural dynamics (Draelos et al., 2021; Churchland et al., 2012; Ablin et al., 2019), additionally develop a new method for streaming dimensionality reduction, and consider multiple possible manifold representations in parallel due to the streaming nature of our algorithm. Finally, we present a novel optimization problem to design high-dimensional stimulations that are aligned to specified desired movements in the low-dimensional space. The problem is constrained by the number of neurons or channels to target and by the non-negative magnitude of total stimulation applied, to simulate realistic experimental conditions. In this step, we leverage the differentiability of our stimulus-response mapping to design stimuli that can adapt to the idiosyncrasies of any individual experiment.

We test using simulated and real neural data across two types of modalities: faster datarates with intracortical electrophysiological recordings and slower datarates with calcium fluorescence activity traces. We design and test multiple kinds of relevant stimulations in the latent subspace, with various constraints on the dimensionality of the resultant stimulation vector. The constraints accommodate realistic experimental conditions, where neurons can be individually addressed yet the number of simultaneous targets and/or the total amount of power is limited (Fernandez-Ruiz et al., 2022; Telliez et al., 2025). Our stimulation targets include the direction of highest neural variance (the first principal component), random feasible directions, and arbitrary (possibly partially infeasible) directions in the latent space. Our algorithms were able to quickly learn a stimulation-response mapping within roughly 10-20 total stimulations delivered, and kept end-to-end runtimes at less than 10 ms on average (and below 100 ms) to ensure real-time feasibility. We anticipate that our adaptive method will enable the next generation of experiments capable of designing and testing stimulations of latent neural dynamics in real time, for both basic neuroscience and brain-machine interface applications.

## 2 METHODS

We give an overall procedure for the use of our framework in Algorithm 1. As neural data is acquired, it is dimension reduced to a latent space. A dynamical model is used to track ongoing latent neural dynamics within that low dimensional space. If a stimulation is delivered, we update a response mapping of its effect on the neural dynamics. If a new stimulation is desired, we solve an optimization problem to determine the closest feasible stimulation in the neural data space to result in a selected perturbation direction in the latent space. All components are updated at each time point in a steaming manner.

---

**Algorithm 1** Real-time Adaptive Stimulation Framework

---

1: **Given:** Neural data stream $\{y_t\}$, latent space mapping $\mathcal{Q}$, dynamical model $f$, stimulus-response model $\hat{S}$, response delay $d$.
2: **Returns:** Optimized stimulus $u^*$ at decision timepoints
3: **Initialize:** Set $t \leftarrow 0$, $\hat{S} \leftarrow \emptyset$, stimulus history $\mathcal{H} \leftarrow \emptyset$
4:
5: **for** $t = 1 \dots$ **do**
6:     Observe new neural data $y_t \in \mathbb{R}^N$
7:     Update latent projections: $x_t \leftarrow \mathcal{Q}.\text{update}(y_t)$       ▷ Observe and project to latent space
8:     $\hat{x}_{t+1} \leftarrow f(x_t)$       ▷ Predict next latent state
9:     **if** stimulation delivered at time $t - d$ (i.e., $(t - d, u_{t-d}) \in \mathcal{H}$) **then**
10:         $s_{\text{obs}} \leftarrow x_t - \hat{x}_t$       ▷ Compute observed response
11:         $\hat{S} \leftarrow \hat{S}.\text{add}(x_{t-d}, u_{t-d}, s_{\text{obs}}, t)$       ▷ Update kernel regression
12:     **else**
13:         $f \leftarrow f.\text{update}(x_t, x_{t-1})$       ▷ Update dynamics model
14:     **end if**
15:     **if** new stimulation desired at time $t$ **then**
16:         **Given:** target direction $v \in \mathbb{R}^k$
17:         $\mathcal{L}(u) = -\frac{v^\top s(u)}{\|v\|\|s(u)\|} + \lambda_1(\|u\|_0^{\max} - \|u\|_1)$       ▷ Optimization problem in (8)
18:         where $s(u) = \hat{S}(x_t, u, t)$       ▷ Predicted response via learned mapping
19:         $u^* \leftarrow \text{argmin}_{u \in [0,1]^N} \mathcal{L}(u)$       ▷ Solve with box constraints
20:         Deliver stimulation $u^*$ to neural system
21:         Add $(t, u^*)$ to $\mathcal{H}$       ▷ Track pending stimulation
22:     **end if**
23: **end for**

---

## 2.1 STREAMING CONSTRUCTION OF LATENT SPACES

Designing and adapting to stimulations in a dynamic latent space first requires that such low-dimensional representations be available in real time. There are multiple hypotheses for which kind of representation might best describe the underlying computation implemented by the brain; for example, highest-variance latent dimensions (Draelos et al., 2021), latents with rotational structure (Churchland et al., 2012), or maximally statistically independent latents (Ablin et al., 2019). Here, we propose a novel streaming latent space construction method, use it alongside two existing methods, and demonstrate that all algorithms are stable approximations of their offline counterparts.

**Novel streaming method.** jPCA (Churchland et al., 2012) is a widely used subspace identification method that identifies planes (pairs of dimensions) with high rotational structure. It achieves this by solving for the best skew-symmetric linear dynamical system that describes the data, based on a comparison of the low-dimensional neural state $X$ with its time derivative $\dot{X}$:

$$M_t = \underset{M}{\text{argmin}} \left\| \dot{X}_t - X_{t-1}M \right\|_2^2 \qquad \text{s.t. } M = -M^\top \tag{1}$$

A dimensionality-reduction step is required to first transform the data into a latent space; (Churchland et al., 2012) used PCA and here we use proSVD (Draelos et al., 2021) as it provides real-time estimates. We then implemented a solution to equation (1) using the Sherman-Morrison formula. jPCA makes a basis out of $M_t$'s eigenvectors: $U_t \Sigma_t U_t^\top = M_t$. To stabilize the subspace, we added a new Orthogonal Procrustes step to stabilize each discovered plane of rotation independently:

$$U_{t,i} = (U_t)_{[2i\,:\,2i+1]} \tag{2}$$

$$\Omega_{t,i} = \underset{\Omega^\top \Omega = I}{\text{argmin}} \left\| (U_{t,i})\Omega - \tilde{U}_{t-1,i} \right\|$$

$$\forall i,\ \tilde{U}_{t,i} = (U_{t,i})\Omega_{t,i}$$

Our novel streaming formulation, named sjPCA, allows us to iteratively estimate a jPCA space in real time that quickly identifies the same space as a later offline calculation.

**Comparison with existing methods**. We compared the above method with two existing methods: proSVD and mmICA. proSVD is a fast, stable, online dimensionality reduction method. It uses an iterative factorization $Y \approx QRW^\top$ of the high-dimensional data $Y$ to learn a set of low-dimensional subspace vectors whose columns form $Q$, and an Orthogonal Procrustes minimization of the change in bases across time. proSVD seeks to track the highest variance $k$-subspace over time; when its inputs are centered, this corresponds to the space containing the top $k$ principal components.

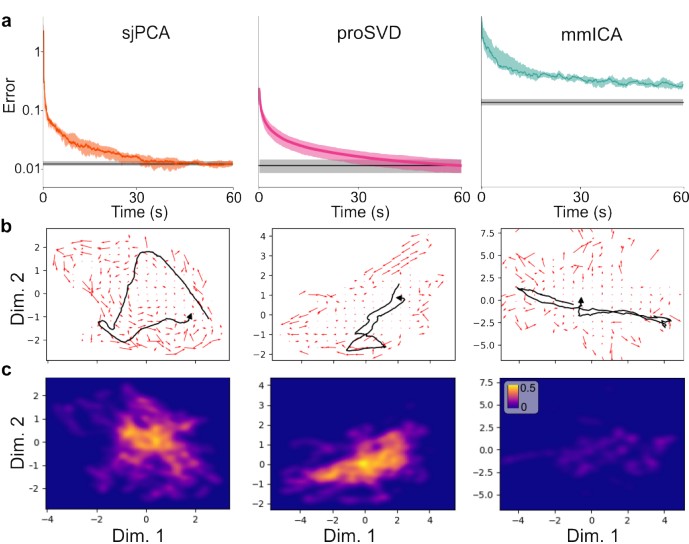

Figure 1: **Real-time manifold construction and dynamical modeling**. **a**. Each streaming dimensionality reduction method (colored lines) converges to a similar representation as one computed offline (black lines). Shaded regions are 1 standard deviation of the errors ($N = 10$ runs). **b**. Projecting real neural data (O'Doherty, 2024) into different latent spaces reveals distinct large-scale dynamical patterns. Quiver plots are the averaged dynamics, with the same 12.5s period of data shown in black. Arrows indicate average direction of flow. **c**. Running the algorithms in parallel allows us to adaptively switch between spaces based on current performance. Heatmaps are estimates of where that space is most likely to give the best predictive probability (modeled using Bubblewrap). Color denotes empirical frequency of being the best predictor across all data.

The two dimensionality reduction methods discussed so far have focused on high variance as a proxy for importance, but other statistical features such as independence may construct better latent spaces. To compare against a non-variance-prioritizing method, we adapted iterative algorithm for independent component analysis (ICA) using a minimization-maximization framework, termed mmICA, that seeks to model input data as linear mixtures of independent components (Ablin et al., 2019). mmICA assumes that the neural data is a linear mixture of independent sources, and uses a maximum likelihood majorization-minimization algorithm to infer the mixture of components that recovers the initial independent sources. Here we apply a proSVD reduction to an initial latent space before using mmICA to learn independent latent dimensions.

**All methods converge to offline fits.** Figure 1a demonstrates convergence to an offline fit. We use a simulated circular linear dynamical system embedded in a latent space for sjPCA and proSVD. mmICA is given a 6D system generated with Laplace random variables where the dimensions are jointly independent, to match the algorithm's assumptions of super-gaussian independent components. Error is measured appropriately for each unique space. For proSVD, we calculate the sum of absolute principle angles between $Q_{:4}$ and the true plane of highest variance. For sjPCA, we similarly compute the sum of the absolute values of the principle angles between the true plane of highest rotation and the identified plane of highest rotation. For mmICA, error is calculated as the Frobenius norm of the difference between the found demixing matrix (normalized with respect to scaling and permutation, see (Ablin et al., 2019)) and the true demixing matrix. Each latent space could be used as a stand-alone space, or considered in parallel to adaptively determine the most useful (predictive) representation at a given timepoint or after certain stimulations are observed.

## 2.2 DYNAMICAL MODELING OF NEURAL LATENTS

We utilize three existing methods for streaming prediction of latent neural dynamics: a simple Kalman filter (KF) (Kalman, 1960), a method based on variational joint filtering (VJF) (Zhao & Park, 2020), and a non-parametric method Bubblewrap (Draelos et al., 2021) that captures probability flow using a joint Gaussian mixture model-hidden Markov model. All models are well suited for

modeling a linear dynamical system, with VJF and Bubblewrap preferred for higher noise regimes or less consistent and multimodal dynamics.

With any of the above dynamical models and latent spaces determined in real time, we can iteratively estimate a flow field that represents the underlying neural manifold discovered by a construction method (Fig. 1b). This gives us the opportunity to compare across latent spaces in parallel and evaluate if there are local regions where the predicted flow field best aligns with newly observed neural data (either spontaneous or evoked via stimulation). All dynamical models in the previous section are evaluated for error in their predictions at every timepoint, allowing us to select from among latent spaces and dynamical models the best performing system at any time. To evaluate the predictive utility of the latent spaces we consider here, we determine the predictive error at each timepoint and aggregate this information within a local region of the latent space (Fig. 1c). Our algorithm thus finds times and locations where each of the spaces yields the best predictions. Such a method could be used, for example, to identify when an animal switches between subtasks with distinct manifold structures (Perkins et al., 2024).

### 2.3 MAPPING DESIRED RESPONSES TO STIMULI

To use stimulation to interrogate neural latents, we need to first characterize how the stimulations affect the latent dynamics. But the mapping between stimuli and neural responses could be non-trivial. There is evidence that responses are driven by network structure and the state of the neural system, and to effectively design stimuli in a real-time setting we need to determine the specific system responses under a wide variety of possible conditions (O'Shea et al., 2022). We do not assume that the response to stimulation is robust nor faithful to the intended stimulation: a neuron may lack sufficient opsin to respond, or the point-spread function is non-optimal and causes out of focus excitation, or other inputs to the neural circuit are active; and thus the response can be different than expected (Ronzitti et al., 2017; Russell et al., 2024; Lees et al., 2024).

**Instantaneous response model.** We first assume a latent dynamical system with the framework:

$$x_{t+1} = f(x_t) + S(x_t, u_t) \cdot \mathbf{1}_{\{u_t \neq 0\}} + \epsilon_t, \tag{3}$$

where $x_t$ is the latent state at time $t$, $f$ is the autonomous mapping of the state from one timepoint to the next, $S$ is a function describing the effect of a stimulation on a location in the latent state, and $\epsilon$ is a noise term. Here, $u$ denotes the stimulation vector itself, potentially quite high-dimensional, and a zero $u$ results in no stimulation and therefore no response affecting the dynamics. Most of the time $u_t$ will be zero, as we are assuming stimulations happen somewhat sparsely on the timescale of the neural data acquisition. This means we can train our estimate of the dynamics, $\hat{f}$, on the datapoints where we know $u_t = 0$, during periods of non-stimulation (details in Appendix A).

$$\hat{f}_{t+1} = \begin{cases} \text{update}(\hat{f}_t, x_{t+1}, x_t), & \text{if } u_t = \mathbf{0} \\ \hat{f}_t, & \text{if } u_t \neq \mathbf{0} \end{cases} \tag{4}$$

To estimate $S$, we can rearrange our dynamics equation: $S(x_t, u_t) = x_{t+1} - f(x_t) - \epsilon_t$ and substitute in $\hat{f}_t$: $S(x_t, u_t) \approx s_{\text{obs}} = x_{t+1} - \hat{f}_t(x_t)$. This gives the following update rule for $\hat{S}$:

$$\hat{S}_{t+1} = \begin{cases} \hat{S}_t, & \text{if } u_t = \mathbf{0} \\ \text{update}(\hat{S}_t, u_t, s_{\text{obs}}, t), & \text{if } u_t \neq \mathbf{0} \end{cases} \tag{5}$$

Together, we can use $\hat{f}$ and $\hat{S}$ to create a joint predictive model:

$$\hat{x}_{t+1} = \hat{f}_t(x_t) + \hat{S}_t(x_t, u_t, t) \tag{6}$$

**Delayed response model.** In many cases, the response to stimulation is not instantaneous, or the peak response to stimulation is not instantaneous. We model these cases using a paradigm similar to the one above, but using a fixed delay $d$: $x_{t+1} = f(x_t) + S(x_{t-d}, u_{t-d}) \cdot \mathbf{1}_{\{u_{t-d} \neq 0\}} + \epsilon_t$. Training of $\hat{f}$ is mostly the same when $d > 0$, except timesteps between a stimulus and its response are left out of training. (Even in steps where the parameters of the $\hat{f}$ estimator is not updated, the estimated state is still tracked.) We assume that a new stimulus is not delivered before we see the effects of a previous stimulus, so that there is never more than one stimulus "pending" at a given time. We

also optionally model the additive effects of stimulation as being spread out over time; if $\hat{x}_{t+1} = \hat{f}_t(x_t) + \hat{S}_t(x_t, u_t, t)$, we optionally regress a small number of coefficients $\beta$ to model the continuing effects of stimulation even after the stimulus is over: $\hat{x}_{t+i+1} = \hat{f}_{t+i}(x_{t+i}) + \beta_i \cdot \hat{S}_t(x_t, u_t, t)$.

**Stimulus-response mapping estimator ($\hat{S}$).** For our model of $S$, we employ a kernel regression to model the effects of latent state, stimulus, and sample age by interpolating between previously observed stimulus-response pairs. We choose radial basis functions for our kernels $K$, where each scaling constant is optionally tuned by stochastic coordinate descent at each new observation.

$$\hat{S}(x, u, t) = \frac{\sum_{i=1}^{N} K_1(x, X_i) K_2(u, U_i) K_3(t, T_i) s_{\text{obs},i}}{\sum_{i=1}^{N} K_1(x, X_i) K_2(u, U_i) K_3(t, T_i)}. \tag{7}$$

Kernel regression works well on limited data (few experimental observations of the results of stimulations), handles possible non-linearities in the response space, and is thus sufficiently flexible for learning potentially non-trivial stimulation-response maps across arbitrary latent spaces (Chen & Shah, 2025). The consideration of sample age ($T_i$) allows it to discount old samples; this means that the regression can tuneably respond to instabilities in the underlying mapping, whether they are due to changes in upstream processing steps or biological changes like plasticity. If the system is stable, it can also use a very large radial basis scaling constant to effectively ignore the time feature.

## 2.4 Optimization of selected stimulations

Stimulations can be designed using a variety of methods: some are based on anatomical region (Shang et al., 2024), on functional tuning of individual neurons (Draelos et al., 2025), on estimated uncertainty (Draelos & Pearson, 2020), on optimal experimental design to choose between a set of predetermined stimuli (Wagenmaker et al., 2024), or simply via random selection of groups of neurons. Here, instead of choosing from a limited set of predetermined stimuli, our method considers all possible stimuli, presenting a considerably larger space to search for feasible stimulations that nonetheless result in a desired effect on the latent dynamics. The tradeoff for this increased flexibility is a more approximate optimization and solution.

We define a goal vector $v$ in the latent space along which we want to perturb the latent neural activity. We control the stimulus vector $u$, and we model the perturbation it produces as $s$ (which depends on $u$). The goal is choose $u$ to get $s$ to align closely to $v$. Under ideal conditions, the values of $u$ are the same as the responses they evoke $s$, and we can optimize $u$ against $v$ directly. We call this an identity stimulus-response mapping, or open-loop optimization. However, such mappings are often more complicated, which is why we also optionally model the evoked response as $s = \hat{S}(x_t, u, t)$ using the learned stimulus-response mapping (Fig. 2a). This adaptation to nonlinear stimulus-response mappings is possible because of the differentiable form of the estimator we use for $\hat{S}$.

We can only stimulate $N$ neurons, and each neuron must have a stimulation value between $0$ and the maximum possible, which we normalize to $1$. Rather than employ the $L_0$ constraint on the number of neurons, which would make the problem NP hard in general, we use an $L_1$ constraint on $u$ offset by $N$ to encourage a solution with the number of non-zero elements close to $n$.

$$\min_{u \in \mathbb{R}^N} -\frac{v^\top s(u)}{\|v\| \|s(u)\|} + \lambda_1 (\|u\|_0^{\max} - \|u\|_1), \quad \text{s. t.} \quad \mathbf{0} \preceq u \preceq \mathbf{1} \tag{8}$$

# 3 Experiments

All experiments were conducted on custom-built workstations running Ubuntu 22.04 and containing 128 GB of RAM, a 32-core i9 intel CPU, an NVIDIA 3060 Ti GPU (8 GB memory), with a 1TB SSD. Experiments could all be run at high speeds, meaning total computation time was kept to less than 100ms, and averaged less than 10ms end-to-end for each timepoint of observed data.

**Toy model.** Our toy model is a circular linear dynamical system defined using: $x_t = Ax_{t-1} + \epsilon_t$, $y_t = Cx_t + \eta_t$, where $A$ is a rotation matrix in the first two components with a period of $30 + \frac{1}{\pi}$ ($\frac{1}{\pi}$ is added to discourage point clustering in adjacent rotations) and a decay to zero in the third component. $C$ is an identity matrix, and $\epsilon_t$ and $\eta_t$ are process and observation noises respectively, both distributed as $\mathcal{N}(0, I_3 \cdot 0.05)$. The initial state $x_0$ is typically initialized to $[20\,0\,0]^\top$.

Stimulations are a binary decision at each timpoint; variation in stimulation magnitude and direction is due to the spatial structure of the stimulation-response mapping, $S$. In the toy model, $S$ is:

$$S_\theta(x_t, u_t) = \begin{cases} \begin{bmatrix} 0 & 0 & 0 \end{bmatrix}^\top, & \text{if } u_t = 0 \text{ or } (x_1 = 0 \text{ and } x_2 = 0) \\ \begin{bmatrix} 0 & 0 & \frac{10(\cos(\theta)x_1 - \sin(\theta)x_2)}{\sqrt{x_1^2 + x_2^2}} \end{bmatrix}^\top, & \text{if } u_t = 1 \end{cases} \quad (9)$$

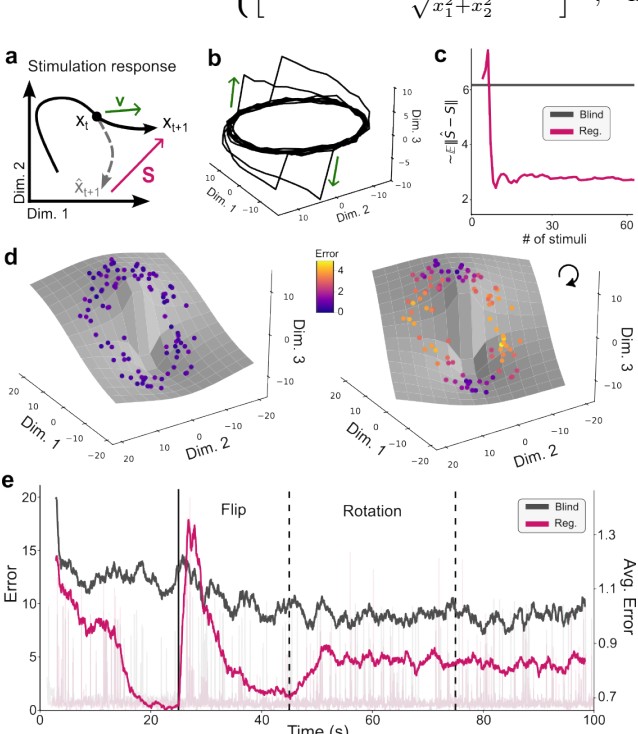

**Real data.** For each of the real datasets, we simulated stimulations using an autoregressive function to model a fast rise in neural activity of the perturbed neurons and a slower decay back to baseline levels. We transformed the data using the following updates: $y_t = r_t + a_t$, $a_t = 0.8 \cdot a_{t-1} + u_t$, where $r_t$ and $y_t$ are the original and simulated data, respectively, $a_i$ is the additive stimulation, and $u_t$ is the stimulation. Figure 3a illustrates two example stimulations applied to the calcium imaging dataset where only stimulated neurons are displayed.

*Calcium imaging.* For the calcium imaging data, we used calcium traces recorded from mouse visual cortex expressing GCaMP6s (Zong et al., 2022). During the recording, the mice were foraging for dropped cookie crumbs the experimenter periodically threw into the environment. Frames were recorded with a miniscope at $15\,\mathrm{Hz}$, for a recording duration of $20\,\mathrm{min}$. The recordings were analysed with Suite2p (Pachitariu et al., 2016), which extracted 592 neural traces. We de-meaned each channel of the florescence output F, defined $F_0$ as the median of each channel, and performed subsequent analyses on $\frac{\Delta F}{F_0} = \frac{F - F_0}{F_0}$.

*Electrophysiological.* We used an electrophysiological dataset from a nonhuman primate (O'Doherty, 2024), recorded from 130 units in the sensorimotor cortex (monkey I).

Figure 2: **a**. Diagram of a trajectory (black) whose dynamics are predicted to advance via the dashed trajectory (gray). If a stimulation $v$ occurs, the activity instead proceeds along the new trajectory. $S$ shows the latent response to stimulation. **b**. A circular system with location-dependent perturbation effects, showing 10 cycles (black). Stimuli displaces along the third dimension (red arrows). **c**. Expected norm-error between our estimate $\hat{S}$ and the true generating $S$ over time. **d**. Surface plots showing the ground-truth effect of stimulations (left: stable, right: rotating). Scattered points are previously observed stimulus-response examples, colored by error. **e**. Error in the 1-step-ahead prediction for our regression method (magenta) and a comparative method that is blind to stimulation effects (gray). The underlying stimulus-response function changes (vertical lines), but the model adapts its temporal kernel length constant to recover.

During the recording, the animal was performing a 2D random-touch task. Threshold crossings were extracted from a $24.4\,\mathrm{kHz}$ recording and binned at $30\,\mathrm{Hz}$ over a recording length of $649\,\mathrm{s}$.

## 4 RESULTS

### 4.1 STIMULATION RESPONSE MODELING

We first applied our response mapping method to the toy model (Fig. 2). Our regression estimator $\hat{S}$ quickly learns the underlying mapping function $S$ within a few seconds, or cycles, of the circular dynamics being observed. To model the kind of instabilities found in real experiments, we first introduced a jump discontinuity, such as when an electrophysiological probe's position is shifted.

To model such a discontinuity in the ground-truth stimulus-response mapping, we flipped the map $180°$ at t=25s (Fig. 2d). While a non-adaptive model that assumes a stable mapping would suffer increased predictive error after such an event, our model recovers from the perturbation within 15s (Fig 2e, 'Flip'). A second kind of instability we considered is continuous drift, which could be caused by photobleaching, plasticity, or a change in neuromodulator levels. To model drift in the ground-truth stimulus-response mapping, we continuously rotated the stimulus-response mapping at a rate of 1 revolution every $30$ s, starting at $t = 45$ s. Our model continuously adjusts to mitigate the error in estimating the unstable underlying system (Fig 2e, 'Rotate'). We quantify the error in 1-step-ahead prediction across all timepoints for our method as well as for a method that is blind to the stimulation by withholding stimulation times from the dynamical model. Both methods employed the same underlying dynamical model (KF) and their errors were similar during periods of non-stimulation. During and post stimulation, our method out-performed the blind comparison method (bold lines show smoothed average errors over 50 experiments).

We next considered real experimental data from (Zong et al., 2022) with simulated stimulations applied along the first latent dimension $Q_0$ as constructed in real time by proSVD (Fig. 3). We confirmed that the applied stimulations had the intended effect on the neural data in both the original high-dimensional neural space and in the learned latent space (Fig. 3a, b, respectively). In real experiments, there is often a lag between stimulus delivery and response, so we introduced a response delay of $0.2$ s (or 4 timepoints) at $t = 304$ s. For both regimes, the one-step-ahead prediction error from our model is less than the error from the blind model. For this dataset, we used the KF method as the dynamical model (see Appendix C for comparison across all models). In all cases, our method quickly learned a stimulation-response mapping to account for the effects of stimulations in the latent space, and out-performed the comparison method.

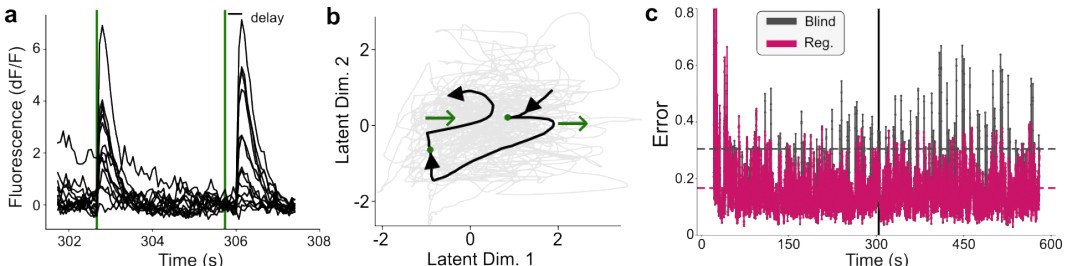

Figure 3: **a**. We apply a stimulus to 14 out of 592 neurons at timepoints $302.6$ s (with a delay of $0$ s) and $305.7$ s (with a delay of $0.26$ s). The new fluorescence traces (black) show a varied effect on activity post stimulation (green vertical line). **b**. The delivered stimuli have the desired effect of pushing the neural trajectory (black) along the first latent dimension $Q_0$ in the latent space constructed with proSVD (rightwards; green arrows). **c**. We plot the 1-step-ahead prediction error as a function of time and dynamic stimulations. Our model (magenta) successfully learns the response to stimulations, whereas the blind model (gray) consistently shows greater error during periods of stimulation. Dashed lines show respective averages during stimulations.

## 4.2 STIMULATION OPTIMIZATION

Previous neuroscience experiments have delivered optogenetic stimuli, though none used strategies for stimulating along latent directions. We can asses the degree to which a stimulation had the desired effect by checking the angle between $v$, the effect of stimulation we desired, and $s_{\text{obs}}$, the deviation from previously predicted dynamics. First we tried stimulating random individual neurons. We found that the effect of activating random neurons had generally low alignment with our desired result of $Q_0$. We then tried maximally activating groups of random neurons; this also did not align well with $Q_0$. We then found that using the stimulations found with our method produces responses highly aligned with $Q_0$ in the latent space, while shuffled versions of our stimulations do not. Via four comparisons, we found that our optimization outperforms random methods in designing stimuli that produce our desired latent effects.

We showed above that we can stimulate along the first dimension in the latent space. However, our system also needs to be able to design stimuli to move neural latents in arbitrary directions. Therefore, next, we quantified how well we can target perturbations in arbitrary directions in the latent space by comparing the $s(u)$ from equation (8) to $v$.

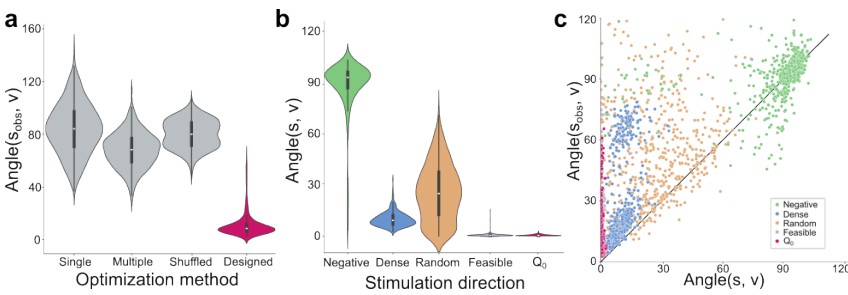

Figure 4: **a**. Random methods, such as randomly stimulating single neurons (Single), groups of neurons (Multiple), or randomized versions of the stimuli our method designs (Shuffled), all produce stimulations that are less aligned with our target effect than the optimized stimuli (Designed). **b**. The predicted angle between the responses we expect ($s$) and the desired response ($v$) for the designed stimuli. We compare the results of optimizations for population-wide inhibition (Negative), population-wide excitation (Dense), random directions in the latent space (Random), random directions constrained to be feasible (Feasible), and along the first latent ($Q_0$). **c**. Observed stimulation error (angle between $s_{\mathrm{obs}}$ and $v$) plotted against predicted stimulation error (angle between $s$ and $v$). Predicted error functions as a loose lower bound on the observed error.

This quantifies how well the optimization predicts it was able to design the stimulus. First, we considered stimulating in an infeasible direction, equivalent to requesting blanket inhibition $v \propto -Q^{\top}\mathbf{1}$ (Fig. 4, 'Negative'). Due to our nonnegativity constraint, any effect of stimulation we design could not possibly be correlated with $v$, just like how it is complicated to optogenetically inhibit activity by targeting excitatory opsins expressed in an excitatory neural population (Li et al., 2019). As expected, we see the angles between the designed $s$ from the optimization and the infeasible $v$ were high. We next checked the performance against another infeasible direction, blanket excitation $v \propto Q^{\top}\mathbf{1}$ ('Dense'). This is similar to blanket excitation that can be delivered by traditional optogenetic manipulations. While inhibition is infeasible due to our non-negativity constraint, blanket excitation is infeasible due to our sparsity constraint. Third, we optimized to stimulate along random directions in the latent space ('Random'). The wide distribution of angles suggests that while some directions are easy for the optimization to target, others are not. We then optimized to stimulate along random feasible directions in the latent space, where we designed the requested vectors to be reachable using the excitation of fewer than 30 neurons ('Feasible'). This case had the best performance, with $517/600$ optimizations giving an optimization misalignment of less than $1°$. Finally, we checked optimizing stimulations to push the population activity along the first latent variable, $Q_0$, which we found to be similarly easy, with $508/600$ optimizations giving an optimization misalignment of less than $1°$.

Another way our stimulations could be challenged is if we have a poor understanding of the mapping from a stimulus to the neural response. So far we have compared the angle between the predicted result of stimulation, $s$, and $v$ and the estimated result of stimulation, $s_{\mathrm{obs}}$ and $v$. We next quantified how these estimates of our error correspond to each other. If we predict based on our optimization that the effects of our stimulation will have a certain error, we should expect the result of the stimulation to have at least that error. If we compare the angle between $s$ and $v$, the predicted error, with the angle between $s_{\mathrm{obs}}$ and $v$, the observed error, we can see that for a variety of targets, the true angle between $s_{\mathrm{obs}}$ and $v$ is greater than the predicted error. For non-'Negative' targeted stimulations, fewer than $6\%$ of optimizations had a lower observed error than predicted. This relationship holds the least for optimizations for the Negative target,

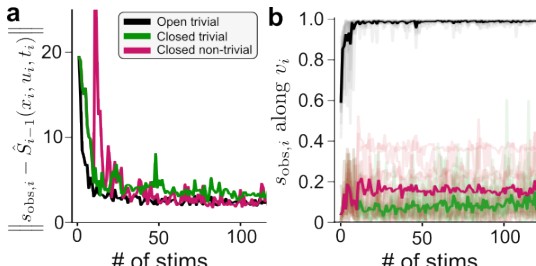

Figure 5: **a**. For each experiment, $\hat{S}$ captures more of the true structure of $S$ over time and has lower prediction error on new training samples, confirming the convergence of $\hat{S}$ as a standalone estimator. **b**. Proportion of the magnitude of the observed $s_{\mathrm{obs}}$ aligned with $v$ for the open loop cases under trivial and non-trivial mappings, and for the closed-loop case for non-trivial maps. 10 experiments are run with over 100 stimulations each; solid lines are average values across experiments.

where about half of optimizations ($296/600$) have a lower error than the optimization predicted, possibly because its infeasibility led our model to predict the maximum possible error.

What if we had seen disagreement between $s_{\text{obs}}$ and $s$? This would indicate our $\hat{S}$ model is a poor match for the system's true $S$. The above experiments assumed that the result of a stimulation $u$ was simply its projection into the latent space $S(u) = Q^\top u$. Because this requires no feedback, or information about the result of the stimulation, we call it open loop mode. In closed loop mode, we can assume a more general form for $S$, but it must be learned via our $\hat{S}$ estimator in real time. Using such an estimated $\hat{S}$, we can see in Fig. 5a that $\hat{S}$ learns at approximately the same rate in a simple (black) vs. non-simple (green) stimulus-response mapping (final error values for individual trials were overlapping: $2.21 \pm .9$ for the simple mapping and $1.95 \pm 0.79$ for the non-simple mapping (mean $\pm$ std). This is because our $\hat{S}$ estimator is non-parametric and makes few assumptions about the underlying stimulus-response mapping. Thus the simple mapping is about as easy to learn as the non-simple mapping. If we then use this estimator to optimize in the non-trivial stimulus-response mapping case, we find that on average, the stimuli designed through the model have a larger proportion of their magnitude aligned with $v$ than the open-loop stimuli (see Appendix G for an analysis of angles in these experiments).

## 5 DISCUSSION

In this work, we describe a new streaming algorithm for stimulation-response modeling of latent neural dynamics, along with a novel optimization procedure for determining high-dimensional stimulation patterns to drive them in a desired direction. This provides, for the first time, a method for adaptive stimulation of latent neural activity that accounts for realistic experimental constraints in the original neural space. Importantly, we considered non-negative constraints for excitation-only interventions, a limit on the number of total targets in a single stimulation, and constraints on the overall magnitude of the applied stimulation. Our optimization framework operated in both the high- and low-dimensional spaces appropriate for this problem of driving latent dynamics via high-dimensional neural stimulations under feasibility constraints. We demonstrated our method's capabilities on synthetic data and two real experimental datasets with simulated effects of arbitrary stimulations, applied both in and out of the learned spaces.

One limitation of our demonstrated approach is that we did not explicitly test using non-linear methods to construct the latent spaces. However, we note that this component of our method could be exchanged without affecting the other components (e.g., using kernelized PCA (Schölkopf et al., 1997) for dimension reduction). A second limitation is that our real data experiments were performed offline, though in a realistic streaming setting. All aspects of our approach run efficiently and are fast enough to make real-time adaptive stimulation experiments feasible (see benchmarking in the Supplementary Materials. We also did not include any explicit consideration of the effects of stimulations on behavior. We note that a straightforward extension of our response-modeling method would be to (separately or jointly) model changes in a lower-dimensional behavioral space. This is feasible for many motor-relevant experiments in neuroscience, as in a 2-dimensional maze or reaching task, or via projecting behavior to its own latent representation (Stringer et al., 2019; Sani et al., 2021; Schneider et al., 2023). Future work could also include additional feasibility constraints on the nature of the stimulation; for example, targeting neurons with more opsin for photostimulation or based on their functional response properties to external stimuli (Russell et al., 2024; Draelos et al., 2025; Daie et al., 2021).

## 6 ETHICS STATEMENT

The authors are not aware of any potential violations of the ICLR Code of Ethics. We do not use human data, we only use publicly available datasets. We do not expect any harm to come from this work's methodologies, insights, or feasible applications. We are not aware of any conflicts of interest. We are not aware of any research integrity issues.

## 7 REPRODUCIBILITY STATEMENT

We will make all code necessary to reproduce our work publicly available via an installable Python package and repository on Github. Additionally, the code behind our method and the code to generate all of the figures in this document is available in the Supplementary Material.

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

## A    ADDITIONAL DETAILS FOR DYNAMICAL MODELS

All three of our dynamical models to predict neural trajectories are Bayesian filters. This means that, given observations $X_1 \ldots X_t$, our models not only can produce a prediction about where $X_{t+T}$ will be but also define a predictive distribution $f(x) = p(X_{t+T} = x)$. We can use this property to validate our predictive methods by comparing their predictions on a known dynamical system. Here, we used the linear dynamical system defined in the main text (without stimulations) where the KF was trained for 10 rotations, BW was trained for 250 rotations, and VJF was trained for 500 rotations. At the end of training, we recorded the next observation from the linear dynamical system, $X_a$, as well as the 0-step predictive distribution and the half-rotation predictive distribution ($p_{a \to a}$ and $p_{a \to b}$). We allowed both the linear dynamical system and the inference to proceed for another half-rotation, and recorded $X_b$ (and the corresponding the 0-step predictive distribution and half-rotation predictive distribution ($p_{b \to b}$ and $p_{b \to a}$).

For all dynamical systems, we checked that the following inequalities held:

$$p_{a \to a}(X_b) < p_{a \to a}(X_a) \tag{10}$$
$$p_{a \to b}(X_b) > p_{a \to b}(X_a)$$

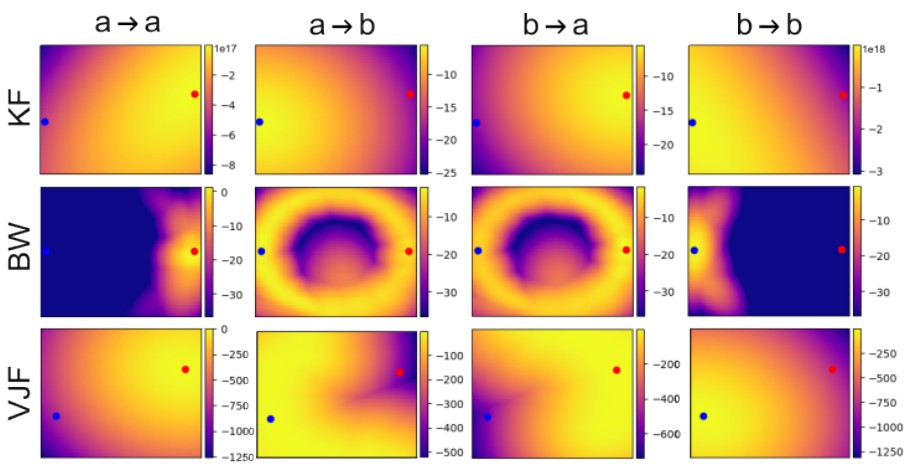

Figure A.1:  Half-turn predictive distributions for the Kalman filter (KF), Bubblewrap (BW), and VJF predictive methods we used to model latent dynamics. Locations for a and b are marked with red and blue dots, respectively.

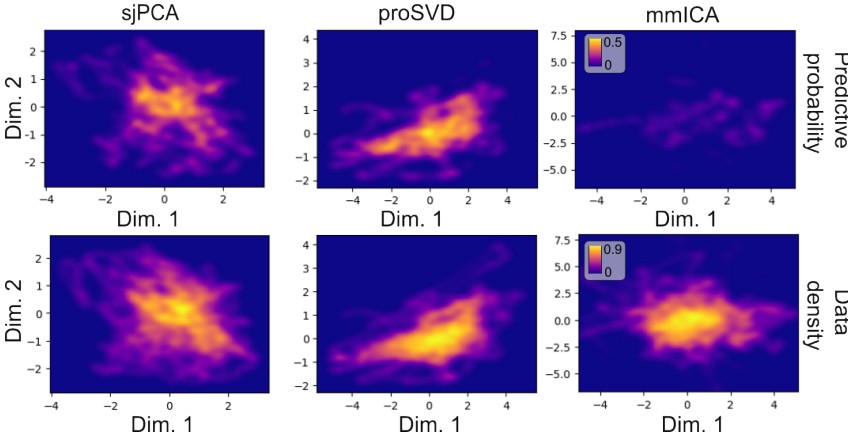

Figure A.2:  Probability distributions (top) reproduced from Figure 1 compared with a heatmap of data densities (bottom) for each latent space.

We looked at how much the centers of the learned spaces changed over time for both the (O'Doherty, 2024) electrophysiological neural data and the (Draelos, 2025) calcium fluorescence neural data. We quantified both the relative change in the centers as well as the relative change in the covariances. For the first dataset, we found that the stepwise differences in the estimate of the mean are less than .5% of the magnitude of the total mean for all timepoints after 16s, and the frobenius norms of the stepwise differences in the estimate of the covariance matrix of the full-dimensional data are less then 1% of the final covariance estimate for all times after 88s. For the second dataset, we similarly found that differences in the center were down to 0.5% after 113s, and covariances down to 1% after 690s. Overall we would consider these quite stable.

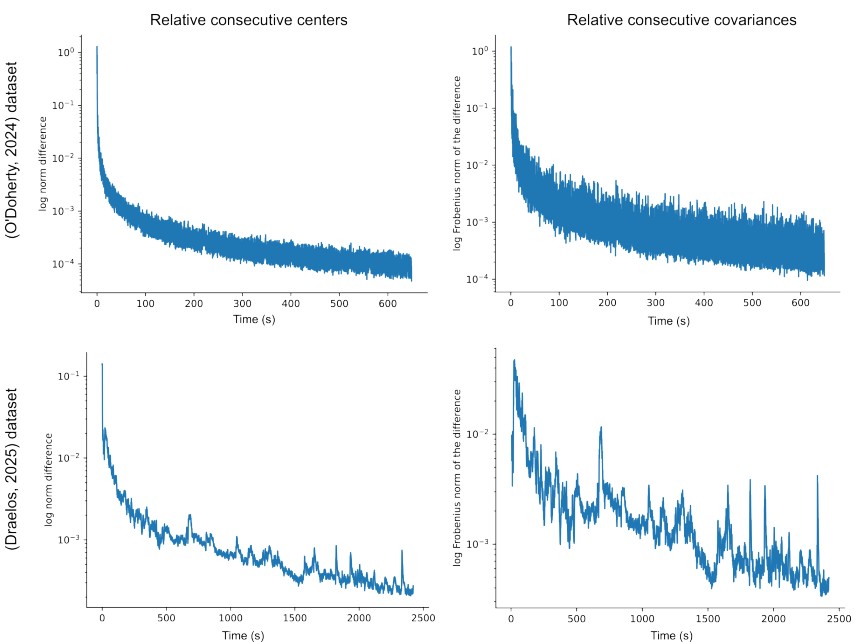

Figure A.3: Stability in online centering with proSVD for two different datasets.

Because these all latent space construction methods are streaming methods, it is straightforward to run them in parallel, or even switch among them to leverage the best predictor at any given timestep. Here we show how at each timepoint a different representation might be used as the best predictor. Using such an adaptive selection of spaces improved average log predictive probability from -1.72 with the best space (proSVD) to -1.01 using all three spaces. We anticipate that this parallelization and streaming capabilities would enable future experiments where best-modeled responses lie in a single latent space, and stimulations are conducted to disambiguate candidate spaces.

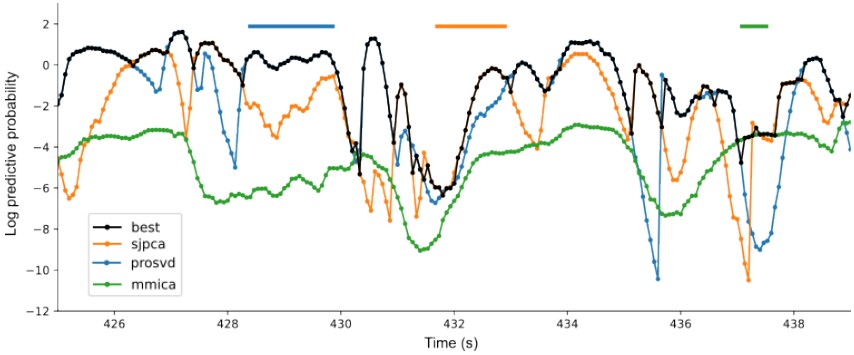

Figure A.4: Example of best predictive probability when adaptively switching between spaces. Manually highlighted regions at the top are where one space is consistently used for a time.

## B  STIMULATION RESPONSE ESTIMATION UNDER OTHER DYNAMICAL MODELS

While we use a Kalman filter as the predictive algorithm for many of the main figures for simplicity, our stimulus regression and correction framework also works on the other dynamical models we consider; namely, Bubblewrap and VJF. The performance of our stimulus regression and correction is somewhat dependent on the performance of the underlying dynamics prediction model. Thus in cases where the underlying prediction model is mismatched to either the neural dynamics or the stimulation effects, our method may perform worse than the blind model.

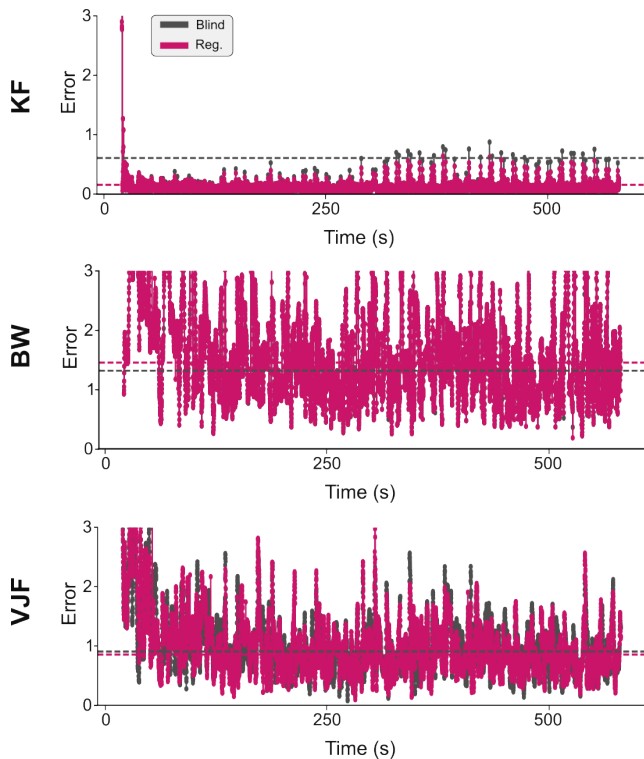

Figure B:  1-step prediction error for a Kalman filter (KF), Bubblewrap (BW), and VJF models. All models were run on the data from Zong et al. (2022) that had been de-meaned, smoothed, and dimension-reduced with proSVD, all in a streaming manner.

## C  STIMULATION RESPONSE ESTIMATION DURING REAL STIMULATIONS

We examine two datasets that conducted direct neuronal stimulations and provide neural time traces and stimulation events that we used for additional validation of our stimulus-response regression model.

The first dataset contains two-photon calcium fluorescence traces during photostimulation of motor cortex neurons in mice during a memory guided response task Daie et al. (2021). The experiment consisted of photostimulations with three different patterns of small groups of neurons (fewer than 10). As the data are discontinuous trials of single stimulation events, we randomly interleaved them to create a continuous data stream and then proceeded to learn a stimulus-response map with our method. Our regression model obtained an average prediction error of 1.84 compared to 2.63 in the blind comparison case (lower is better; similar to Figure 3c).

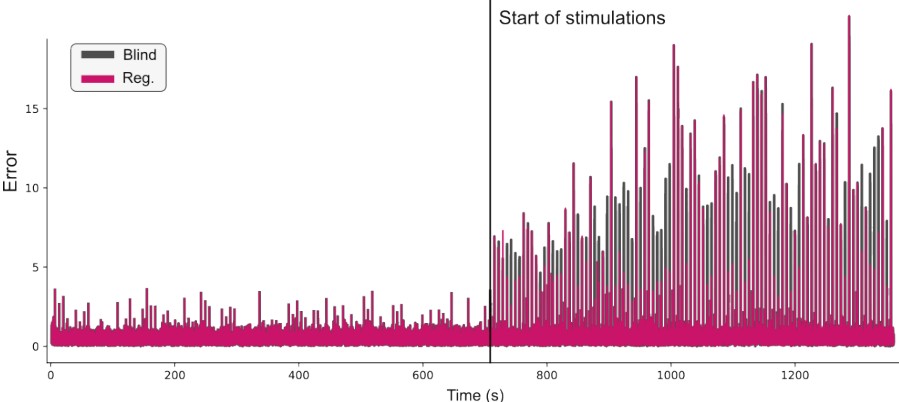

Figure C.1: 1-step prediction error for our regression model on the stimulation dataset from Daie et al. (2021). Our method averages 1.84 error compared to 2.63 with the blind comparison method during the photostimulation period.

The second dataset contains two-photon calcium fluorescence traces during photostimulation of head-fixed larval zebrafish Draelos et al. (2025). The experiment consisted of photostimulations with different targets and was continuous across time. Our regression model obtained an average prediction error of 4.63 compared to 5.67 in the comparison case.

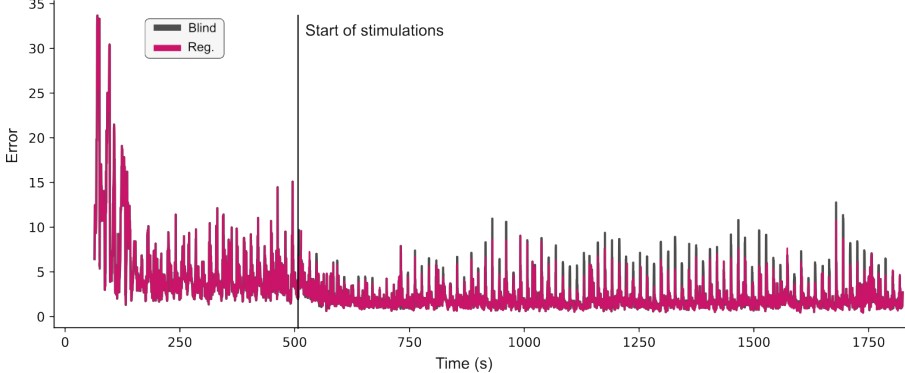

Figure C.2: 1-step prediction error for our regression model on the stimulation dataset from Draelos et al. (2025). Our method averages 4.63 error compared to 5.67 with the blind comparison method during the photostimulation period.

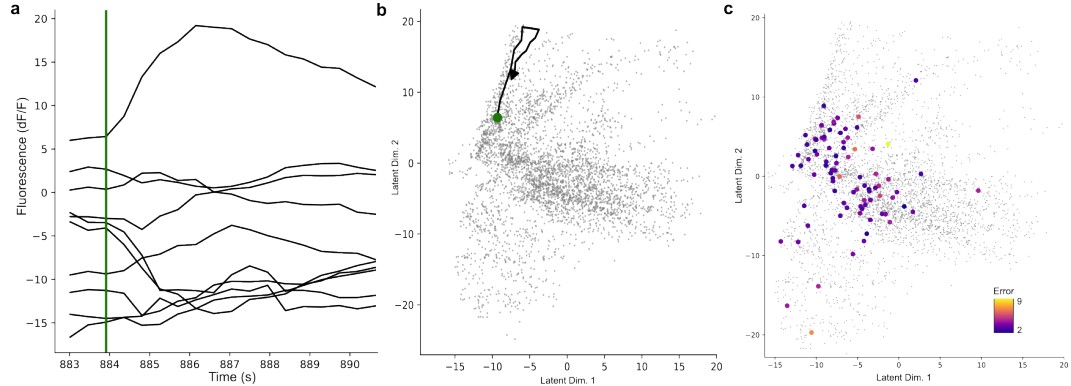

Figure C.3: **a.** Change in fluorescence traces as a function of time post-stimulation event. **b.** Example of one neural trajectory post-stimulation event in the first two dimensions of the latent space (proSVD). **c.** Scatter points showing the error in stimulation prediction overlaid across the same latent space.

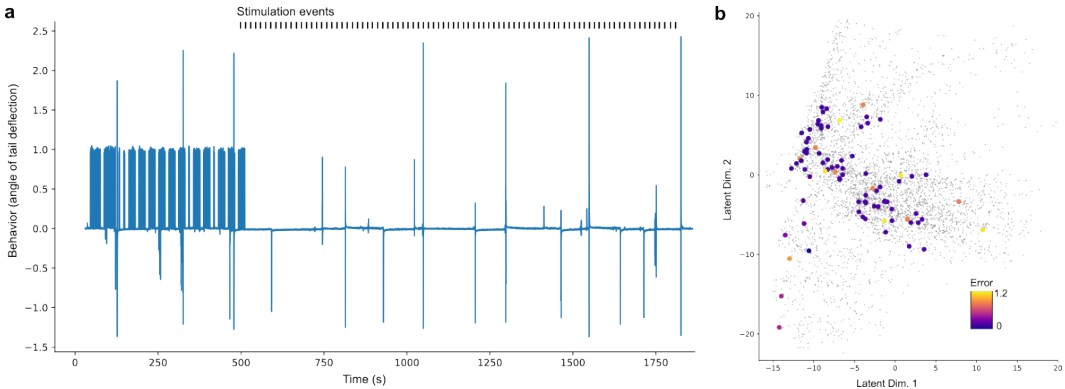

Figure C.4: **a.** Tail angle deflection as behavior plotted as a function of time through the dataset. Photostimulation events begin where marked, and behavior is noticeably sparser than earlier in the trial, where visual stimuli were presented. **b.** Scatter points showing the error in stimulation prediction of behavior overlaid across the neural latent space.

## D  OPTIMIZATION IN A TOY MODEL

Our stimulus regression attempts to account for non-trivial stimulus-response relationships. Designing stimuli using our kernel-regressed stimulus-response map trades computational complexity for more general stimulus design. This means that, in the absence of a complicated stimulus-response tradeoff, our model will overfit and negatively impact performance. To investigate this tradeoff, we compared simulations in our toy dataset between trivial vs. non-trivial stimulus-response maps and open vs. closed loop estimators.

Simulations were conducted with either a trivial $S(x, u)$ mapping (consisting of just dimensionality reduction) or a non-trivial $S(x, u)$ (consisting of dimensionality reduction and a permutation). For each of these simulation types, we trained two estimators. One estimator, the open-loop estimator, assumes the trivial $S(x, u)$ mapping, while the other estimator, the closed-loop estimator, learns $S(x, u)$ using our new kernel regression method. (In these simulations, $S$ ignores its $x$ input.)

One hypothesis is that the best performance would correspond to the open-loop estimator on the trivial stimulus-response mapping, because the open-loop estimator's prior exactly matches the simulation's simple stimulus-response mapping. We also anticipated that the worst performance would

correspond to the open-loop estimator on the non-trivial stimulus-response mapping, because the open-loop estimator's prior would not match the non-trivial stimulus-response mapping. This would leave the two closed-loop estimators in the middle, performing worse than a correct prior but better than an incorrect prior. We did not predict for there to be a systematic difference between the closed-loop simulations, although they may occur because the different stimulus-response mappings would mean that the two closed-loop estimators are targeting different directions in the latent space, and as we show in Figure 4a in the main text, targeting the first proSVD latent appears to be the easiest.

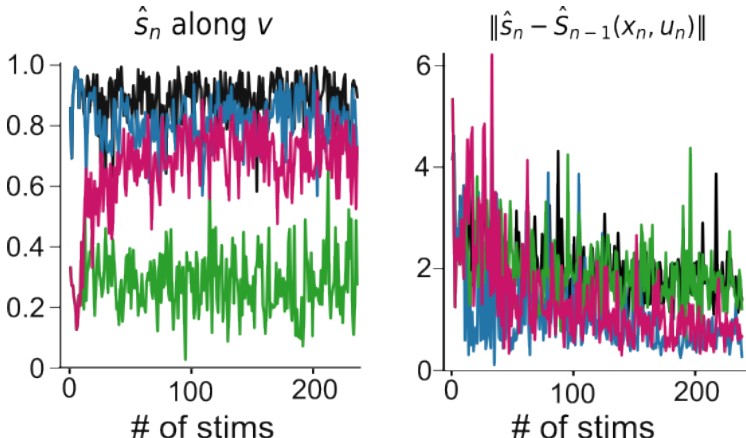

Figure D.1: Open vs closed loop stimulus optimization on the toy dataset. The black trace corresponds to the trivial simulation and open-loop estimator, the blue trace corresponds to the trivial simulation and closed-loop estimator, the green trace corresponds to the non-trivial simulation and open-loop estimator, and the magenta trace corresponds to the non-trivial simulation and closed-loop estimator. The first 10 stimuli of each trial were open-loop in order to initialize the kernel regression. All traces are an average of 10 trials.

Next, we also tested more complex stimulations in this toy model and the ability of our regression method to account for them, similar to Figure 2 in the main text. Both models perform slightly worse than in the simple 1-dimensional stimulation case, but our regression model still well outperforms the blind comparison method. Our regression method obtained an average error of $1.01 \pm 0.30$ compared to $1.33 \pm 0.42$ in the blind comparison method during the last rotational block (as in the last portion of Figure 2e).

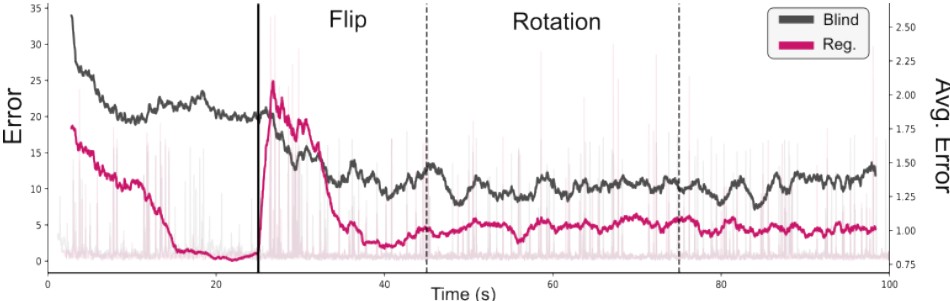

Figure D.2: Error in the 1-step-ahead prediction for our regression method (magenta) and a comparative method that is blind to stimulation effects (gray) during stimulations in all three dimensions of the toy model.

Lastly, we also implemented a toy model with non-rotational linear dynamics in the x-y plane, where the ground truth response map is similar to the one in Figure 2 with stimulations displaced along the z direction. We projected this 3-dimensional data into a higher 8-dimensional space, and then used sjPCA for dimension reduction. We observed that the data thus projected into the top

two dimensions of sjPCA space did not appear overly rotational, but mimicked the structure of the original 3-dimensional data.

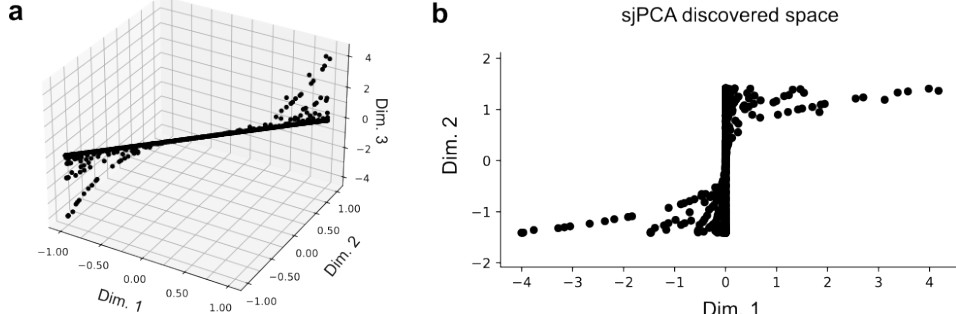

Figure D.3: **a.** Toy model with non-rotational dynamics where stimulations are along the third dimension. **b.** Data projected back onto the first two dimensions discovered by sjPCA after original data was projected into an 8-dimensional space.

## E  OPTIMIZING STIMULATIONS THROUGH VARIOUS LATENT SPACES

While the main paper only shows stimulus optimization in a latent space identified using the proSVD method, our stimulation algorithm is capable of designing stimuli in all latent spaces we considered in the main text. In Figure 4a of the main text, we show that our stimulus optimization is sensitive to alignment with the first latent variable discovered by proSVD. This could be because proSVD discovers latent variables that are easier to optimize for (proSVD's first latent variable is often the highest variance), or because when we were developing the method we tuned the optimization's parameters using the alignment of stimulations with the first proSVD latent as a metric. In either case, sjPCA and mmICA reorganize the latent variables discovered by proSVD, which would lead us to expect optimization results like those in the $\hat{r}$ case in Figure 4a. However, the low performance of the closed-loop optimization on the sjPCA latents in the trivial stimulus-response mapping simulation is worse; further exploring this quirk may give insight into the space discovered by sjPCA or our closed-loop optimization.

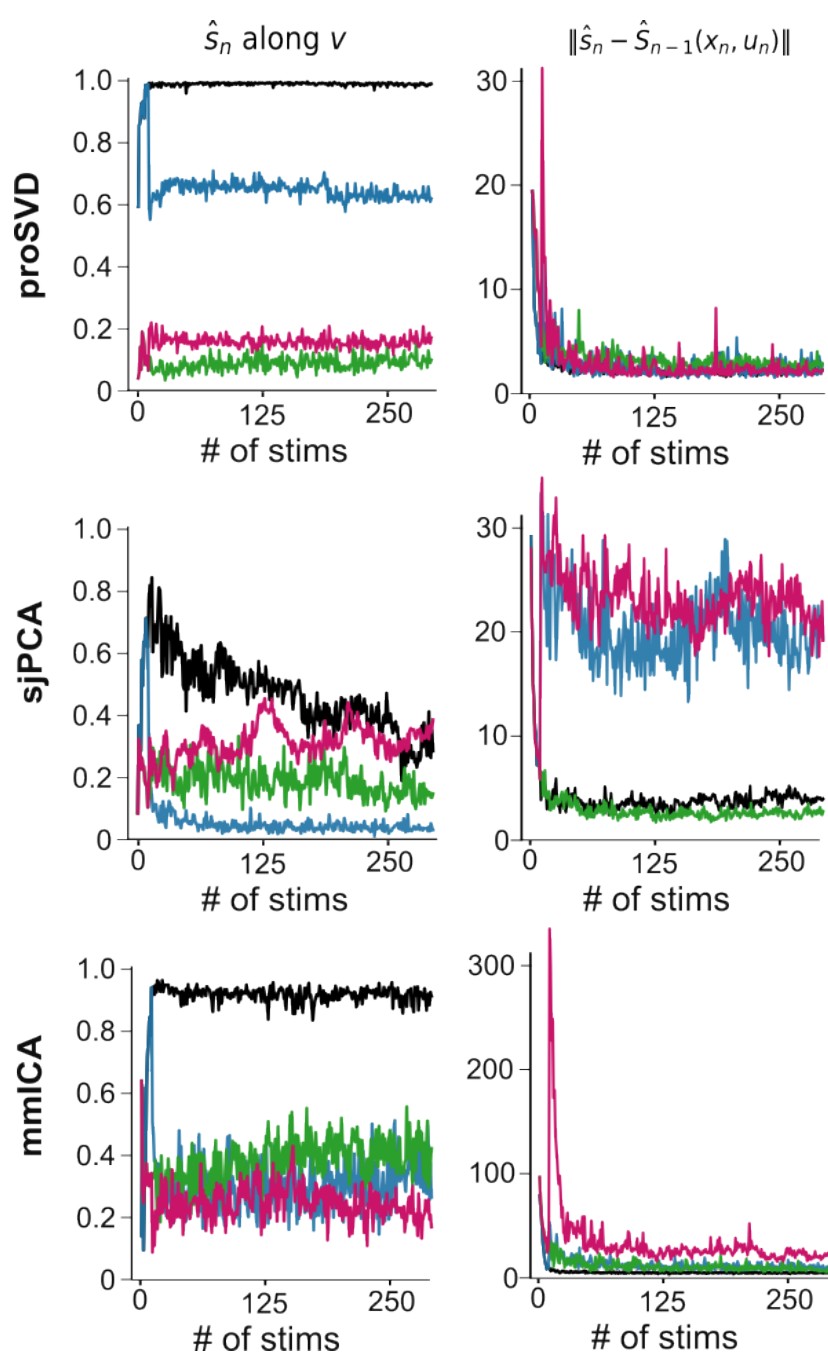

Figure E: Stimulus optimization performance for proSVD, sjPCA, and mmICA. Stimuli were designed during a simulated experiment based on real experimental data O'Doherty (2024). Stimuli were designed to activate the first latent variable identified by proSVD, and were delivered at random times at a rate of about 1 stimulation every 2 seconds. Left plots quantify the alignment between the goal stimulus ($v$) and the change in dynamics the stimulus regression model observed ($\hat{s}_n$). Right plots quantify predictive error on new observations ($\hat{s}_n$) in the stimulus response regression ($\hat{S}$).

Across all datasets and experiments (open and closed-loop), we found that the proSVD latent space yielded the best results on average, independent of dynamical model. However, there were certain conditions we tested (open-loop, flipped response map) where sjPCA provided the best latent space representation. We also found that the Kalman filter provided the best results in the majority of the

cases studied here. The tables below list all results for all spaces and dynamical models used for the two datasets in the main text.

Table 1: odoherty21, kf, s_hat

| Condition | prosvd | sjpca | mmica |
|---|---|---|---|
| open | $2.25 \pm 0.36$ | $3.04 \pm 0.63$ | $4.53 \pm 0.58$ |
| closed | $10.21 \pm 1.19$ | $13.07 \pm 1.57$ | $25.46 \pm 3.10$ |
| open flip | $3.17 \pm 0.67$ | $2.50 \pm 0.54$ | $7.74 \pm 2.82$ |
| closed flip | $10.49 \pm 1.28$ | $12.97 \pm 1.49$ | $24.96 \pm 3.82$ |

Table 2: odoherty21, bw, s_hat

| Condition | prosvd | sjpca | mmica |
|---|---|---|---|
| open | $2.48 \pm 0.44$ | $3.19 \pm 0.68$ | $4.83 \pm 0.66$ |
| closed | $10.51 \pm 1.27$ | $13.37 \pm 1.62$ | $25.94 \pm 3.19$ |
| open flip | $3.30 \pm 0.79$ | $2.61 \pm 0.65$ | $7.93 \pm 2.80$ |
| closed flip | $10.76 \pm 1.38$ | $13.31 \pm 1.74$ | $25.78 \pm 4.21$ |

Table 3: odoherty21, vjf, s_hat

| Condition | prosvd | sjpca | mmica |
|---|---|---|---|
| open | $2.40 \pm 0.41$ | $3.15 \pm 0.66$ | $4.84 \pm 0.72$ |
| closed | $10.46 \pm 1.31$ | $13.34 \pm 1.61$ | $25.86 \pm 3.34$ |
| open flip | $3.23 \pm 0.75$ | $2.55 \pm 0.55$ | $7.87 \pm 2.87$ |
| closed flip | $10.73 \pm 1.35$ | $13.27 \pm 1.59$ | $25.58 \pm 4.22$ |

Table 4: odoherty21, kf, 1step_pred

| Condition | prosvd | sjpca | mmica |
|---|---|---|---|
| open | $1.83 \pm 0.52$ | $1.65 \pm 0.46$ | $4.00 \pm 1.04$ |
| closed | $1.54 \pm 0.59$ | $1.60 \pm 0.72$ | $3.36 \pm 1.43$ |
| open flip | $1.80 \pm 0.51$ | $1.75 \pm 0.49$ | $3.91 \pm 1.05$ |
| closed flip | $1.56 \pm 0.61$ | $1.62 \pm 0.74$ | $2.92 \pm 1.43$ |

Table 5: odoherty21, bw, 1step_pred

| Condition | prosvd | sjpca | mmica |
|---|---|---|---|
| open | $2.09 \pm 0.68$ | $1.79 \pm 0.58$ | $4.37 \pm 1.19$ |
| closed | $2.12 \pm 1.09$ | $2.24 \pm 1.20$ | $4.23 \pm 2.28$ |
| open flip | $2.02 \pm 0.69$ | $1.86 \pm 0.57$ | $4.17 \pm 1.22$ |
| closed flip | $2.18 \pm 1.12$ | $2.32 \pm 1.26$ | $4.12 \pm 2.49$ |

Table 6: odoherty21, vjf, 1step_pred

| Condition | prosvd | sjpca | mmica |
|---|---|---|---|
| open | $1.96 \pm 0.57$ | $1.70 \pm 0.51$ | $4.33 \pm 1.10$ |
| closed | $1.99 \pm 1.06$ | $2.05 \pm 1.15$ | $4.09 \pm 2.14$ |
| open flip | $1.89 \pm 0.56$ | $1.80 \pm 0.51$ | $4.01 \pm 1.13$ |
| closed flip | $2.01 \pm 1.06$ | $2.11 \pm 1.22$ | $3.92 \pm 2.34$ |

Table 7: zong22, kf, s_hat

| Condition | prosvd | sjpca | mmica |
|---|---|---|---|
| open | $2.04 \pm 0.40$ | $2.46 \pm 0.58$ | $3.12 \pm 0.48$ |
| closed | $7.17 \pm 1.20$ | $8.07 \pm 0.96$ | $13.70 \pm 2.00$ |
| open flip | $1.90 \pm 0.36$ | $2.02 \pm 0.45$ | $5.18 \pm 1.53$ |
| closed flip | $7.23 \pm 0.87$ | $8.14 \pm 0.97$ | $14.46 \pm 2.02$ |

Table 8: zong22, kf, 1step_pred

| Condition | prosvd | sjpca | mmica |
|---|---|---|---|
| open | $1.44 \pm 0.31$ | $1.49 \pm 0.32$ | $2.29 \pm 0.47$ |
| closed | $1.61 \pm 0.56$ | $1.65 \pm 0.62$ | $2.57 \pm 1.03$ |
| open flip | $1.39 \pm 0.31$ | $1.42 \pm 0.31$ | $2.49 \pm 0.55$ |
| closed flip | $1.60 \pm 0.56$ | $1.63 \pm 0.62$ | $2.23 \pm 1.13$ |

Table 9: zong22, bw, s_hat

| Condition | prosvd | sjpca | mmica |
|---|---|---|---|
| open | $3.51 \pm 0.69$ | $3.49 \pm 0.79$ | $4.13 \pm 0.66$ |
| closed | $8.21 \pm 1.48$ | $9.00 \pm 1.23$ | $14.47 \pm 2.25$ |
| open flip | $3.07 \pm 0.55$ | $3.20 \pm 0.68$ | $6.33 \pm 1.55$ |
| closed flip | $8.23 \pm 1.08$ | $8.94 \pm 1.17$ | $15.34 \pm 2.32$ |

Table 10: zong22, bw, 1step_pred

| Condition | prosvd | sjpca | mmica |
|---|---|---|---|
| open | $3.26 \pm 1.15$ | $2.96 \pm 1.20$ | $3.88 \pm 1.15$ |
| closed | $3.50 \pm 1.35$ | $3.31 \pm 1.26$ | $4.20 \pm 1.67$ |
| open flip | $2.73 \pm 0.99$ | $2.73 \pm 1.09$ | $3.91 \pm 1.11$ |
| closed flip | $3.67 \pm 1.56$ | $3.42 \pm 1.34$ | $4.22 \pm 1.85$ |

Table 11: zong22, vjf, s_hat

| Condition | prosvd | sjpca | mmica |
|---|---|---|---|
| open | $2.60 \pm 0.44$ | $2.83 \pm 0.63$ | $3.96 \pm 0.68$ |
| closed | $7.64 \pm 1.70$ | $8.50 \pm 1.08$ | $14.24 \pm 2.31$ |
| open flip | $2.30 \pm 0.40$ | $2.42 \pm 0.49$ | $5.84 \pm 1.57$ |
| closed flip | $7.60 \pm 0.97$ | $8.56 \pm 1.25$ | $15.11 \pm 2.33$ |

Table 12: zong22, vjf, 1step_pred

| Condition | prosvd | sjpca | mmica |
|---|---|---|---|
| open | $2.02 \pm 0.48$ | $1.96 \pm 0.48$ | $3.21 \pm 0.80$ |
| closed | $2.25 \pm 0.89$ | $2.31 \pm 0.91$ | $3.67 \pm 1.48$ |
| open flip | $1.82 \pm 0.44$ | $1.84 \pm 0.46$ | $3.25 \pm 0.84$ |
| closed flip | $2.25 \pm 0.85$ | $2.35 \pm 0.94$ | $3.55 \pm 1.66$ |

## F    OPTIMIZATION IN A SUBSPACE

Consider our optimization in the main text (Equation 8):

$$\min_{u \in \mathbb{R}^N} -\frac{v^\top s(u)}{\|s(u)\|\|v\|} + \lambda_1(\|u\|_0^{\max} - \|u\|_1), \quad \text{s. t.} \quad \mathbf{0} \preceq u \preceq \mathbf{1} \tag{11}$$

We also considered a modification which would allow us to more flexibly optimize for stimuli. If we restrict $\|v\| = 1$, $\|s(u)\| = 1$, and $v^\top s(u) > 0$ the first term is equivalent to

$$-\left\|v^\top s(u)\right\|^2 \tag{12}$$

which is a useful format because it is defined when $v$ is both a vector and a matrix. In the case when $v$ is a vector, this term ensures that we maximize the projection of $s(u)$ in the direction of $v$. When $v$ is a matrix, minimizing this term means maximizing the projection of $s(u)$ along the subspace defined by $v$. Thus not only can our optimization be framed for finding a stimulation aligned with one direction, but for aligning our stimuli with an arbitrary linear subspace. This could be useful when we are experimentally interested in multiple directions at once; passing in a matrix $v$ would allow the algorithm to possibly optimize against the most favorable direction in the subspace spanned by $v$.

## G    ANALYSIS OF CLOSED-LOOP OPTIMIZATION VECTORS

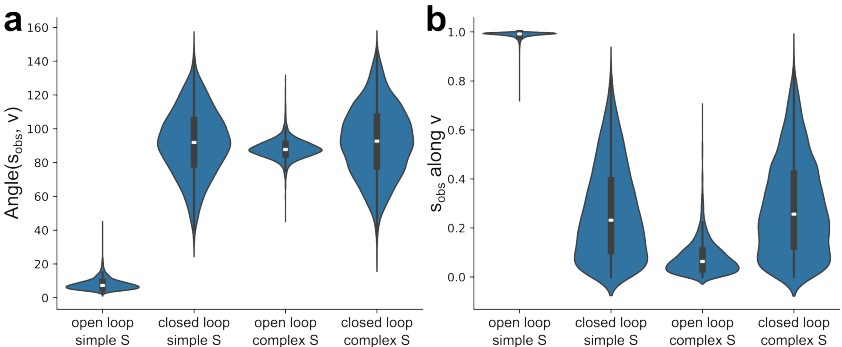

Figure G.1:   Comparison of the angle and projection distance metrics for closed-loop stimulation optimization results (same data as Figure 5). **a**. Angles between $s_{\text{obs}}$ and $v$, like reported in Fig. 4a. **b**. Projection norms of $s_{\text{obs}}$ along $v$, like reported in Fig. 4b.

The higher projection distance of $s_{\text{obs}}$ along $v$ we see for the closed-loop optimization in Fig. 5b appears to be largely driven by an increased angle variance in the results of the optimization. For our two metrics, we have:

$$\texttt{projection\_along}(s_{\text{obs}}, v) = \frac{\left\|v^\top s_{\text{obs}}\right\|}{\|s_{\text{obs}}\|}$$

$$\texttt{angle}(s_{\text{obs}}, v) = \theta = \cos^{-1}\left(\frac{v^\top s_{\text{obs}}}{\|s_{\text{obs}}\|}\right)$$

$$\texttt{projection\_along}(s_{\text{obs}}, v) = |\cos(\texttt{angle}((s_{\text{obs}}, v))|$$

(We chose this form for the `projection_along` metric so it would be compatible with optimization in a subspace, see section F.) We can see in panel a that the distributions of angles between $s_{\text{obs}}$ and $v$ is centered near $90°$. $\cos(90°) = 0$, so it would be reasonable to expect the distributions of projection lengths of $s_{\text{obs}}$ along $v$ in panel b to also be centered at zero, but the norm we use in the projection metric prevents this. If we have $\mathbb{E}[\Theta_1] = \mathbb{E}[\Theta_2] = 0$, and $\mathbb{V}[\Theta_2] > \mathbb{V}[\Theta_1]$, then the expectations of the absolute values will be different: $E[|\Theta_2|] > E[|\Theta_1|]$.

Next, we analyzed how well the optimization is able to perform in the absence of all constraints. We replicated the experiment in Figure 4b but relaxed both the sparsity and non-negativity constraints. We found that indeed now dense, and random stimulations are all now considered feasible and are successfully found by our stimulation optimization algorithm. The negative (as in inhibitory stimulations) have more diverse results, which we attribute to the infeasibility (regardless of optimization constraint) for obtaining negative neural activity.

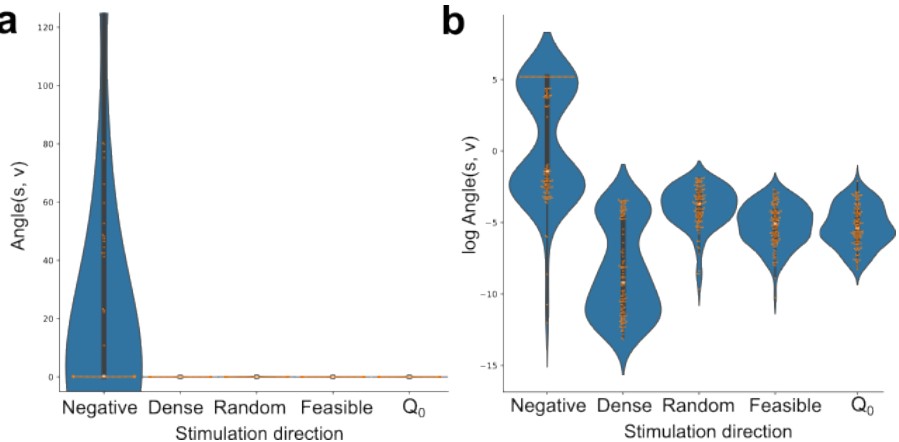

Figure G.2: Repeated experiment to Fig. 4 but without constraints, plotted with the same scale as earlier (**a**) and log-scale to better visualize how all designed stimulations except Negative ones are extremely close to the target angle (**b**).

We next examined the relationship between feasibility and the designed stimulations. For the (Zong, 2021) dataset used in Figure 3, we simulated different radial directions of requested stimulation targets at a single point in the latent space. In some cases, as in directions towards the right, no feasible stimulations exist. In the case of the top left direction, only weak deviations of the neural trajectories are predicted to be possible.

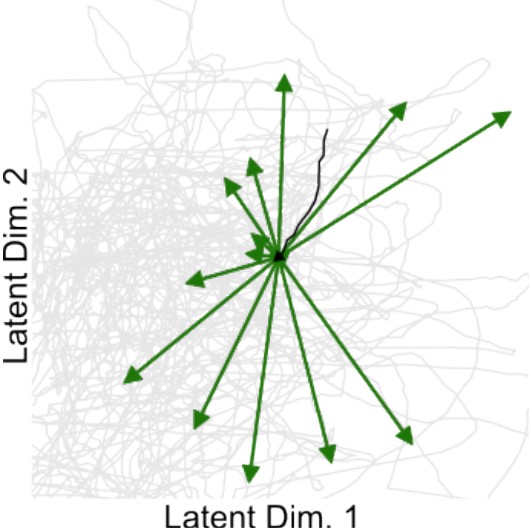

Figure G.3: Target vectors are requested in all directions from a single point in the latent space. Green arrows show the observed simulated responses of the optimized stimulations.

Finally, we analyzed the interaction between direction of motion of ongoing latent neural dynamics and the target vector, and how that affected the outcome of stimulations. We used the O'Doherty

(2024) dataset from main Figure 4, generated stimulations at different timepoints, and calculated the angle between the current direction of motion in the latent space and the desired perturbation ('Dynamics angle'), and the angle between the actually observed response and the desired perturbation ('Response angle'). We observed mostly what we expected; that for a wide variety of possible dynamics angles, if the stimulation was feasible, the response angle was small (well aligned with the target) regardless of ongoing motion. We did not conclude that there was a statistically significant correlation between the angle between ongoing dynamics and the desired target, and the angle between the observed response and the desired target (Feasible: p=0.79, $Q_0 : p = 0.83$).

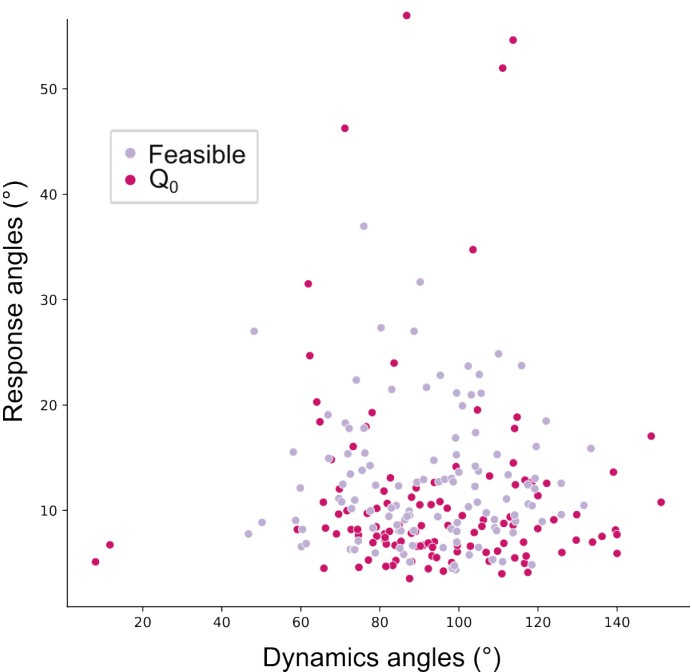

Figure G.4: Target vectors are requested in all directions from a single point in the latent space. Green arrows show the observed simulated responses of the optimized stimulations.

## H BENCHMARKING

### H.1 END-TO-END OPTIMIZATION TIMING

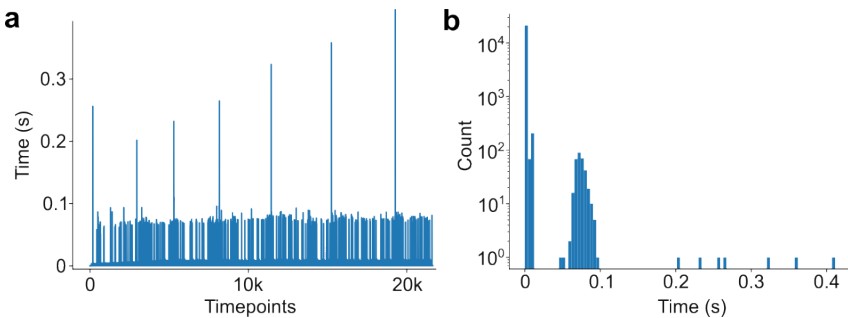

Figure F.1: **a.** Execution times for each step in the end-to-end optimization framework as a function of the number of timepoints through an experiment. **b.** Histogram showing that most execution steps took less than 100 ms.

## H.2   DIMENSIONALITY REDUCTION TIMING

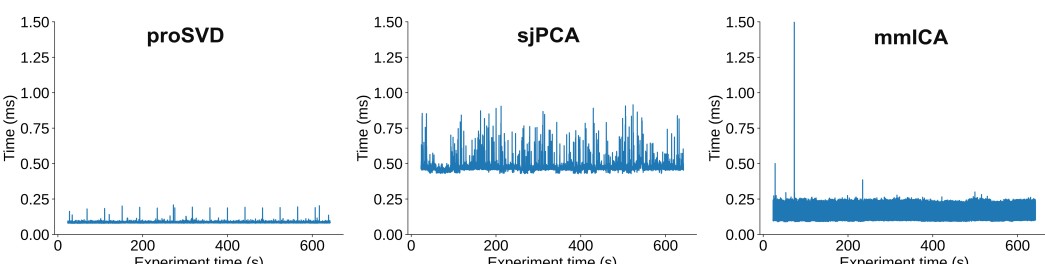

Figure F.2:   Dimension reduction benchmarking for proSVD, sjPCA, and mmICA running on neural data O'Doherty (2024).  All algorithms are executed in $< 2$ ms per step, making real-time space construction feasible.

## H.3   COMPLEXITY

For the stimulation response model, we use a form of kernel regression that minimizes loss in polynomial time Cesa-Bianchi et al. (2015) since we learn a set of coefficients fairly efficiently. For the online or streaming setting, complexity via gradient descent is O(dN) where d is the dimensionality of the space Li & Liao (2023). Here, since d is small and N itself grows relatively slowly (at the rate of stimulation events), this is easily managed.

For the stimulation design algorithm, we implemented the constrained optimization problem with the L-BFGS-B method (which is the limited memory and box-constrained variant of BFGS) Zhu et al. (1997) which has complexity O(iN) where i is the number of iterations and N is the dimensionality of the high-dimensional stimulation vector u (typically, the number of neurons). This will not grow as a function of time throughout an experiment, and so should be constant for a given neural population.

# I  VARIABLES

| Symbol | Shape | Meaning | Algorithm |
|---|---|---|---|
| $t$ | $\mathbb{N}_+$ | number of timepoints recorded so far | |
| $N$ | $\mathbb{N}_+$ | neural data recording dimensionality | |
| $k$ | $\mathbb{N}_+$ | the dimensionality of the low-d space | |
| $X_{t-1}$ | $\mathbb{R}^{(t-1 \times k)}$ | low-dimensional neural data | sjPCA (1) |
| $\dot{X}_t$ | $\mathbb{R}^{(t-1 \times k)}$ | low-dimensional neural data time differences | sjPCA (1) |
| $M$ | $\mathbb{R}^{(k \times k)}$ | rotational linear dynamics (skew-symmetric) | sjPCA (1) |
| $\tilde{U}_{t,i}$ | $\mathbb{R}^{(k \times 2)}$ | $i$th stabilized plane of $U_t$ | sjPCA (2) |
| $\Omega$ | $\mathbb{R}^{(2 \times 2)}$ | Orthogonal Procrustes stabilization rotation | sjPCA (2) |
| $x_t$ | $\mathbb{R}^k$ | latent state at time $t$ | dynamics (3) |
| $u_t$ | $\mathbb{R}^N$ | stimulation vector delivered at time t | dynamics (3) |
| $f, \hat{f}$ | $\mathbb{R}^k \to \mathbb{R}^k$ | autonomous dynamics (and estimated version) | dynamics (3) |
| $S$ | $(\mathbb{R}^k, \mathbb{R}^N) \to \mathbb{R}^k$ | stimulus-response mapping | dynamics (3) |
| $\hat{S}$ | $(\mathbb{R}^k, \mathbb{R}^N, \mathbb{R}) \to \mathbb{R}^k$ | kernel-regressed stimulus-response mapping | regression (7) |
| $X_i$ | $\mathbb{R}^k$ | latent state at $i$th recorded stimulus | regression (7) |
| $U_i$ | $\mathbb{R}^N$ | stimulation delivered at $i$th stimulus | regression (7) |
| $T_i$ | $\mathbb{R}$ | time of $i$th recorded stimulus | regression (7) |
| $s_{\text{obs},i}$ | $\mathbb{R}^k$ | estimated dynamics change due to $i$th stimulus | regression (7) |
| $u$ | $\mathbb{R}^N$ | designed stimulus | optimization (8) |
| $v$ | $\mathbb{R}^k$ | desired dynamics change if $u$ were applied | optimization (8) |
| $s(u)$ | $\mathbb{R}^k$ | predicted dynamics change if $u$ were applied | optimization (8) |
| $\lambda_1$ | $\mathbb{R}$ | $L_1$ regularization constant | optimization (8) |
| $\lVert \cdot \rVert_1$ | $\mathbb{R}^N \to \mathbb{N}_+$ | $L_1$ norm (of $u$) | optimization (8) |
| $\lVert u \rVert_0^{\max}$ | $\mathbb{N}_+$ | maximum acceptable $L_0$ norm of $u$ | optimization (8) |

