# OpenReview forum: "Adaptive stimulation & response modeling of latent neural dynamics"
_ICLR.cc/2026/Conference — Submitted to ICLR 2026_

### Official Review · Reviewer_739W · 2025-10-20

**Soundness:** 3
**Presentation:** 2
**Contribution:** 3
**Rating:** 6
**Confidence:** 3

**Summary:**

This paper proposes a streaming method for stimulation-response modeling. To do that, they propose a new real-time subspace identification method building upon jPCA, a kernel-based approach to model stimulus-response effect, and an optimization problem to find the optimal stimulus to perturb the system along the desired direction. They show the proposed approach can indeed perturb the system in a set of feasible directions both in simulations and real datasets.

**Strengths:**

- The formulation of the sparse-input constraint is well-motivated and seamlessly integrated into the overall framework.
- The proposed streaming subspace identification method achieves performance comparable to proSVD, demonstrating its accuracy and stability in real-time settings.
- Beyond simulations, the approach is also evaluated on real-world datasets, underscoring its potential utility for practical, real-time applications.

**Weaknesses:**

- Can the authors comment on the necessity of a new subspace identification method for their stimulus optimization approach? At each iteration, do they pick the subspace identification method that yields the best performance? How would the results change if they only use sjPCA or proSVD?
- The proposed method is pretty much involved in terms of the required steps, such as subspace identification, dynamic model fitting, stimulus map estimation, prediction, and stimulus optimization (and doing all these iteratively in real time). I think a step-by-step, clear guide on the procedure (starting from a specific time t and perturbing the system in the desired direction at time T) would improve the readability of the paper significantly by helping the reader to connect the dots.
- Do the proposed stimulus optimization provide convergence to a specific location on the latent space, or just perturbation along a specific axis?
- For my own understanding, is it correct that updating the $f_t$ and $S_t$ corresponds to fitting new models?
- The target $v$ is selected based on the identified latent subspace, which does not need to carry a semantic meaning. Without a method to properly identify the map between the latent space and the desired stimulation effect, how can one design the stimulation target? Without such a map, what is the utility of the proposed approach?
- If the sparsity and/or nonnegativity constraint is relaxed, can the authors perturb the system along infeasible directions?
- Can you add a legend for Fig. 5 (in addition to mentioning them in the caption)?
- While computing angles between $s_{obs}$ and $v$, do authors use the $s$ obtained at $t+1$ if the input is applied at $t$?
- Can you add a legend for Fig. 5 (in addition to mentioning them in the caption)?
- It looks like the proposed approach is limited to real-time subspace identification methods. With the increased interest in deep-learning-based foundation models, is it possible to combine the proposed stimulus optimization approach with a pretrained dynamical model? Is it possible to test this in a simulation setup with simple linear dynamical models?

**Questions:**

Please refer to the Weaknesses section.

---

> ### Author Response · Authors · 2025-11-22
>
> We thank the reviewer for their comprehensive feedback, summary of our contributions, and excellent suggestions. We address comments and questions below.
>
> > Subspace methods for stimulation optimization
>
> While we view our core contribution as the closed-loop design of stimulations in the high-dimensional space to produce effects in the latent space, we wanted our framework to be usable in a general way. We thus chose to implement the necessary preprocessing steps of (1) dimension reduction to the latent space and (2) modeling the dynamics of the non-stimulated latent neural activity. Here we mostly leveraged existing methods to show how our method could be integrated with each of them, but also to show that our method is robust to the specific neural representation chosen for an individual experiment.
>
> Because our method is designed for closed-loop and real-time experiments, we also required that preprocessing steps be executable also in real time. We selected proSVD and mmICA as streaming methods that represented the neural data under different assumptions: high-variance directions, and independent components. As jPCA is a common neural representation framework, we wanted to add it as well, and so developed sjPCA as a streaming-capable and time-stabilized implementation.
>
> Because these are all steaming methods, it is straightforward to run them in parallel, or even switch among them to leverage the best predictor at any given timestep. We have added a figure to Appendix A to show how at each timepoint a different representation might be used as the best predictor. We anticipate that this parallelization and streaming capabilities would enable future experiments where best-modeled responses lie in a single latent space, and stimulations are conducted to disambiguate candidate spaces. In practice, and in all experiments presented in the main text, we only utilized one space at a time, as we think this is the most straightforward and likely initial use case by experimentalists.
>
> To compare how results would change by using different spaces, we have now expanded Appendix E and provided results for all sets of experiments run in Figures 3 and 4 (run for each latent space separately, and each dynamical model as well). Across all datasets and experiments (open and closed-loop), we found that the proSVD latent space yielded the best results on average, independent of dynamical model. However, there were certain conditions we tested (open-loop, flipped response map) where sjPCA provided the best latent space representation. We also found that the Kalman filter provided the best results in the majority of the cases studied here. Below we present one example set of results on the (O’Doherty, 2024) dataset used in Figure 4 where we design stimulations using open-loop, open-loop with a flipped mapping, closed-loop, and a closed-loop with flipped mapping between stimulation vectors and responses. A complete listing of results is in Appendix E.
>
> ||s_{obs} – S_{i-1}(xi, ui, ti)||
>
> | Stim. Cond. / Lat. Sp. Model | sjPCA              | proSVD              | mmICA           |
> |------------------------------|--------------------|---------------------|-----------------|
> | Open                         | 3.04 &pm; 0.63     | **2.25** &pm; 0.36  | 4.53 &pm; 0.58  |
> | Open flipped                 | 13.07 &pm; 1.57    | **10.21** &pm; 1.19 | 25.46 &pm; 3.10 |
> | Closed                       | **2.50** &pm; 0.54 | 3.17 &pm; 0.67      | 7.74 &pm; 2.82  |
> | Closed flipped               | 12.97 &pm; 1.49    | **10.49** &pm; 1.28 | 24.96 &pm; 3.82 |

---

> > ### Author Response · Authors · 2025-11-22
> >
> > > Procedure guide
> >
> > This is a great suggestion, thank you. We have added a new algorithm in pseudocode that we hope improves the readability and gives a better overview of the necessary components.
> > We also reproduce it here for easy access.
> >
> > **Algorithm: Real-time Adaptive Stimulation Framework**
> >
> > **Given:** Neural data stream {yₜ}, latent space mapping 𝒬, dynamical model f, stimulus-response model Ŝ, response delay d
> >
> > **Returns:** Optimized stimulus u* at decision timepoints
> >
> > **Initialize:** Set t ← 0, Ŝ ← ∅, stimulus history ℋ ← ∅
> >
> > ---
> >
> > **for** t = 1, 2, ... **do**
> >
> > 1. Observe new neural data yₜ ∈ ℝᴺ
> >
> > 2. Update latent projections: xₜ ← 𝒬.update(yₜ)    &nbsp;&nbsp;&nbsp;&nbsp;&nbsp; > *Observe and project to latent space*
> >
> > 3. x̂ₜ₊₁ ← f(xₜ)     &nbsp;&nbsp;&nbsp;&nbsp;&nbsp;> *Predict next latent state*
> >
> > 4. **if** stimulation delivered at time t−d (i.e., (t−d, uₜ₋ₐ) ∈ ℋ) **then**
> >    - sₒᵦₛ ← xₜ − x̂ₜ     &nbsp;&nbsp;&nbsp;&nbsp;&nbsp;> *Compute observed response*
> >    - Ŝ ← Ŝ.add(xₜ₋ₐ, uₜ₋ₐ, sₒᵦₛ, t)     &nbsp;&nbsp;&nbsp;&nbsp;&nbsp;> *Update kernel regression*
> >
> > 5. **else**
> >    - f ← f.update(xₜ, xₜ₋₁)     &nbsp;&nbsp;&nbsp;&nbsp;&nbsp;> *Update dynamics model*
> >
> > 6. **end if**
> >
> > 7. **if** new stimulation desired at time t **then**
> >    - **Given:** target direction v ∈ ℝᵏ
> >    - Define loss function:
> >
> >      𝓛(u) = −(vᵀs(u))/(‖v‖‖s(u)‖) + λ₁(‖u‖₀ᵐᵃˣ − ‖u‖₁)     &nbsp;&nbsp;&nbsp;&nbsp;&nbsp;> *Optimization problem in (8)*
> >
> >      where s(u) = Ŝ(xₜ, u, t)     &nbsp;&nbsp;&nbsp;&nbsp;&nbsp;> *Predicted response via learned mapping*
> >
> >    - u* ← argmin_{u ∈ [0,1]ᴺ} 𝓛(u)     &nbsp;&nbsp;&nbsp;&nbsp;&nbsp;> *Solve with box constraints*
> >    - Deliver stimulation u* to neural system
> >    - Add (t, u*) to ℋ     &nbsp;&nbsp;&nbsp;&nbsp;&nbsp;> *Track pending stimulation*
> >
> > 8. **end if**
> >
> > **end for**
> >
> >
> > > Convergence or perturbation
> >
> > Great question. Our method is not used for convergence, but is used for perturbation along a given direction in the latent space. We formulated it in this way because we were interested in obtaining a general response map of stimulations across the neural latent space, which could lead to a more principled understanding of dynamics on neural manifolds. This method could in the future be combined with a goal location and an optimization (or control framework) to iteratively perturb latent dynamics until they reached that goal. We note there are other methods designed to solve a similar problem (goal-oriented, not direction or mapping oriented) (Deco, 2019; Tafazoli, 2020; Minai, 2024 & 2025), but there is no guarantee that much else beyond a particular pattern for that one goal location is learned. It would be useful to also provide a way to adapt to changes in responses once a pattern is identified, as our method does. We anticipate in general however that such methods would be complementary to our approach.
> >
> >
> > > Updating f_t and S_t
> >
> > We would consider this to be an iterative updating of an existing model, rather than fitting an entire new model at each timepoint. This means we don’t have to incur computational costs for re-fitting from scratch each time. But it does effectively result in a new model at each timepoint, so you could consider it that way as well.
> >
> >
> > > Semantic meaning of targets
> >
> > This is an interesting question about target selection and the need for a stimulation response map. For target selection, there is indeed not necessarily a semantic meaning to a direction selected in the latent space. We focused on the example of high-variance components (e.g., Q_0) because of its relationship to the dominant neural dynamics in the latent space; a kind of biologically-relevant meaning. But in the general case for our method, the stimulation target can always be designed without a response map. The only thing that is required is the latent space representation.
> >
> > This doesn’t guarantee that the result of the stimulation produces anything close to the target vector, however. As the reviewer points out, a map between the target vector in the latent space and the actual stimulation effect is necessary if stimulations produce any noise, variation, or non-identity responses. This response map is exactly what we learn in section 2.3 to provide this feature as part of the closed-loop stimulation optimization. That closed-loop portion uses the learned stimulation response map to design stimulation patterns that result in motion along the target vector. Without this map, stimulations are indeed less effective (green vs magenta lines, Fig. 5; where we have also added a legend).

---

> > > ### Author Response · Authors · 2025-11-22
> > >
> > > > Relaxation of constraints
> > >
> > > We thank the reviewer for this interesting idea. We have now tried a case where we relax both the sparsity and non-negativity constraints and repeat the experiments conducted in Figure 4b. We found that indeed now dense, and random stimulations are all now considered feasible and are successfully found by our stimulation optimization algorithm. The negative (as in inhibitory stimulations) have more diverse results, which we attribute to the infeasibility (regardless of optimization constraint) for obtaining negative neural activity. This experiment and its results are now in Appendix G.
> > >
> > >
> > > > Legend for Fig. 5
> > >
> > > Thank you for pointing out this oversight on our part; we have added a legend to that figure.
> > >
> > >
> > > > Angle between s_{obs} and v
> > >
> > > Yes; to determine s_{obs} we do wait until time t+1 (or t+delay if a delay is specified) to consider that the actual response to the stimulation.
> > >
> > >
> > > > Deep learning subspace and dynamical models
> > >
> > > This is a great question and a good idea that we had not considered. We could in principle use a dimension reduction method that had been pretrained (or freeze the loadings after a period of training time in real time); this still provides data in the latent space at each timepoint. To run in real time it would need to be sufficiently lightweight, but we find it likely that many neural networks would be capable. Similarly, we could use a pretrained (or frozen) dynamical model of data in the latent space; it simply needs to provide 1 timepoint ahead predictions based on current data. This would be less effective in scenarios where neural dynamics shift over time.
> > >
> > > The harder problem in integration with our stimulation optimization method is in the closed-loop setting. In that case, we optimize with the learned stimulation response map (kernel regression) which we can differentiate through using the streaming dimension reduction methods we used. With a deep learning model of dimension reduction, this may no longer be possible. Perhaps an inverted map might also be learned via a neural network; we think this is definitely a cool idea for future work.
> > >
> > >
> > > **References**
> > >
> > > Deco, G., Cruzat, J., Cabral, J., Tagliazucchi, E., Laufs, H., Logothetis, N. K., & Kringelbach, M. L. (2019). Awakening: Predicting external stimulation to force transitions between different brain states. Proceedings of the National Academy of Sciences, 116(36), 18088-18097.
> > >
> > > Minai, Y., Soldado-Magraner, J., Smith, M., & Yu, B. M. (2024). MiSO: Optimizing brain stimulation to create neural activity states. Advances in Neural Information Processing Systems, 37, 24126-24149.
> > >
> > > Minai, Y., Soldado-Magraner, J., Yu, B. M., & Smith, M. A. (2025). OMiSO: Adaptive optimization of state-dependent brain stimulation to shape neural population states. arXiv preprint arXiv:2507.07858.
> > >
> > > Tafazoli, S., MacDowell, C. J., Che, Z., Letai, K. C., Steinhardt, C. R., & Buschman, T. J. (2020). Learning to control the brain through adaptive closed-loop patterned stimulation. Journal of Neural Engineering, 17(5), 056007.

---

> > > > ### Comment · Reviewer_739W · 2025-11-25
> > > >
> > > > I thank the authors for the detailed rebuttal and additional experiments. The revisions improve clarity, but some of my main concerns remain.
> > > >
> > > > First, as also stated by other reviewers, the subspace-identification component still feels secondary. The new results show that proSVD performs as well as or better than the proposed sjPCA. I acknowledge that authors claim their stimulation design as their core contribution, but the controversy around sjPCA makes the overall novelty of the work less clear.
> > > >
> > > > Second, and more importantly, I still struggle to see the experimental utility of designing stimulations along arbitrary latent directions when these coordinates have no semantic interpretation. Without a meaningful mapping between latent axes and physiological or behavioral variables, the practical impact of “driving dynamics along v” remains unclear. In this regard, the authors’ response suggests that such a mapping from physiological/behavioral variables to the dynamical model's latents can be treated as a black box, but I respectfully disagree. This mapping is crucial for real experimental deployment, and I do not yet see how the proposed framework bridges this gap.
> > > >
> > > > For these reasons, I will keep my original score for now.

---

> > > > > ### Author Response · Authors · 2025-11-27
> > > > >
> > > > > We thank the reviewer for their response. We agree that if the latent space has been designed to carry meaning in its dimensions, that stimulations along those dimensions have a straightforward interpretability.
> > > > >
> > > > > Latent variable models can be also useful without being explicitly designed to relate to external variables. (Yıldız, 2022) proposes a latent space model for seizure identification in EEG. Their latent space was constructed in an unsupervised manner, with no reference to seizure status, and so no single latent variable represented seizure status. Despite this, the latent space was still capable of supporting seizure classification. In a similar way, our method can take an arbitrarily constructed latent space and learn its structure after the fact via stimulations. Figure C.4 in Appendix C shows how our framework could be used to better understand the relationship between neural states and behavior, which we can use to interrogate black-box links. Further work could interrogate whether certain directions of stimulation consistently evoke behavioral responses and the patterns in this relationship.
> > > > >
> > > > > We also find that there are interesting hypotheses that can be tested without reference to external variables. For example, (O’Shea, 2022) proposes a distinction between an activity subspace and dynamics subspace. This difference is between sets of latent dimensions, making no reference to external variables. Their model predicts varying temporal responses to stimulation in sets of orthogonal latent dimensions; our method is capable of designing those stimulations to conduct causal tests of those hypotheses.
> > > > >
> > > > >
> > > > > Finally, we would like to ask what the reviewer means by controversy around sjPCA specifically?
> > > > >
> > > > >
> > > > > (As an aside, when we do want to construct latent spaces with external variables in mind, the streaming PLS algorithm found in `AdaptiveLatents/adaptive_latents/pro_pls.py` in the supplementary material can be used to construct streaming latent spaces with structure linked to an outside variable-like behavior.)
> > > > >
> > > > >
> > > > > **References**
> > > > >
> > > > > O’Shea, D. J., Duncker, L., Goo, W., Sun, X., Vyas, S., Trautmann, E. M., ... & Shenoy, K. V. (2022). Direct neural perturbations reveal a dynamical mechanism for robust computation. bioRxiv, 2022-12.
> > > > >
> > > > > Yıldız, İ., Garner, R., Lai, M., & Duncan, D. (2022). Unsupervised seizure identification on EEG. Computer methods and programs in biomedicine, 215, 106604.
> > > > >
> > > > > Zhou, D., & Wei, X. X. (2020). Learning identifiable and interpretable latent models of high-dimensional neural activity using pi-VAE. Advances in neural information processing systems, 33, 7234-7247.

---

### Official Review · Reviewer_UUNK · 2025-10-31

**Soundness:** 2
**Presentation:** 2
**Contribution:** 2
**Rating:** 2
**Confidence:** 3

**Summary:**

Raw neural data is very high dimensional but actual neural activity patterns that occur during behavior tend to lie on a much smaller, low-dimensional surface within that space which is the neural manifold. The authors give an example of how selecting even just 30 neurons involves searching a space of over 10^45 combinations at least.

The authors explain that prior work has addressed isolated sections of this problem of tracking neural dynamics and designing neural stimulations. This work focuses on bringing down the high dimensionality neural data into low dimensionality latent space in real time (novel streaming method), and then trying to find stimulations that shift this low dimensional space in a particular direction.

Novel streaming method:
First getting the high dimensional data into low dimension through proSVD - the output is a low dimensional state that can update in real time.
jPCA is applied to low dimensional space to figure out the rotational dynamics. doing JPCA will give us the M matrix.
M can drift or be unstable over time. They stabilize the latent subspace (eigenvectors of M) using the orthogonal procrustes step.

They compare their method (sjPCA) to existing streaming dimensionality reduction methods (proSVD and mmICA). All of them converge to similar representations as one computed offline. sjPCA performs on par with the existing methods (and also appears to be slightly faster).

Modeling:
applying stimuli to neural responses is non-trivial and to effectively design stimuli in a real-time setting they want to determine the systems responses under multiple conditions.
For this they create 3 models, each more complex than the previous one. They start with an instantaneous response model assuming that the neural activity responds to the stimulus immediately in the next time step. Next is delayed response model which introduces delay d before the effect appears. Last, a third model that uses Kernel regression to model the effects of latent state, stimulus, and sample age to create a stimulus response mapping estimator.

The authors consider a large space of possible stimuli to search for feasible stimulations. They also acknowledge that the tradeoff with such a large space is that any possible solutions may be approximations. The goal is to find a vector u (stimulus) such that vector s (perturbation) aligns closely to vector v (goal). This is shown in figure 2a.

They try out their methods first on test data, and then real data obtained from calcium imaging with a miniscope. Their model adapts and recovers from externally added perturbation while the non-adaptive model’s error does not decrease.

**Strengths:**

The authors identify a potentially interesting problem of how to make experimental perturbations more efficient and better informed by theory.

**Weaknesses:**

I have several major concerns.  The first is that the premise of this framework is built entirely on jPCA, which is a highly specialized method whose only utility is in finding rotational dynamics.  Contrary to the authors' claims, I am not aware of widespread use of jPCA beyond the original paper itself. Instead, the field has mostly adopted more flexible methods that do not only identify rotational dynamics. I don't understand the authors' motivation for starting with jPCA, as it seems they could have used a more general dynamical systems framework (i.e. no constraint on M) .

**Questions:**

1. Why is jPCA the right starting point, and why is this approach not possible without a constraint on the M matrix?
2. In figure 1, sjPCA does not appear to be outperforming existing methods. Is this a just a matter of clarifying the presentation or is this an accurate assessment  of model performance?
3. How does this method perform on synthetic data with ground truth that is not rotational?

---

> ### Author Response · Authors · 2025-11-22
>
> We thank the reviewer for their feedback and comprehensive summary of our contributions. We address comments and questions below.
>
> > Reliance on jPCA
>
> We want to clarify that the premise of our framework is definitely not built entirely on jPCA, but that this particular latent space representation is one possible representation that our method is able to work with. We agree with the reviewer that other flexible methods that don’t look for rotational dynamics are useful. We thus also demonstrated integration with two others (proSVD, mmICA), and since our framework is not dependent on anything beyond that there exists a latent space and dimension reduction method, we see no fundamental reason it wouldn’t also work with other methods not tested here.
>
> While we view our core contribution as the closed-loop design of stimulations in the high-dimensional space to produce effects in the latent space, we wanted our framework to be usable in a general way. We thus chose to implement the necessary preprocessing steps of (1) dimension reduction to the latent space and (2) modeling the dynamics of the non-stimulated latent neural activity. Here we mostly leveraged existing methods to show how our method could be integrated with each of them, but also to show that our method is robust to the specific neural representation chosen for an individual experiment.
>
> Because our method is designed for closed-loop and real-time experiments, we also required that preprocessing steps be executable also in real time. We selected proSVD and mmICA as streaming methods that represented the neural data under different assumptions: high-variance directions, and independent components. As jPCA is a common neural representation framework, we wanted to add it as well, and so developed sjPCA as a streaming-capable and time-stabilized implementation. While the interpretation of the rotational dynamics learned by jPCA is debatable (e.g, Lebedev, 2019; Kuzmina, 2024), jPCA is widely used in neuroscience (see reviews in Kuzmina, 2023; Perich, 2025; also Rouse, 2018; Awasthi, 2022; Batabyal, 2024; Zhang, 2025), even beyond motor contexts (e.g., prefrontal cortex in Aoi, 2020), and in other ML problems (e.g., evaluating RNNs in Smith, NeurIPS 2021). It is by no means the best latent representation for all data, even motor cortex data, but is a widely used starting point.
>
> We have now expanded Appendix E and provided results for the experiments run in Figures 3 and 4. Across all datasets and experiments (open and closed-loop), we found that the proSVD latent space yielded the best results on average, independent of dynamical model. However, there were certain conditions we tested (open-loop, flipped response map) where sjPCA provided the best latent space representation. We also found that the Kalman filter provided the best results in the majority of the cases studied here.
>
> ||s_{obs} – S_{i-1}(xi, ui, ti)||
>
> | Stim. Cond. / Lat. Sp. Model | sjPCA              | proSVD              | mmICA           |
> |------------------------------|--------------------|---------------------|-----------------|
> | Open                         | 3.04 &pm; 0.63     | **2.25** &pm; 0.36  | 4.53 &pm; 0.58  |
> | Open flipped                 | 13.07 &pm; 1.57    | **10.21** &pm; 1.19 | 25.46 &pm; 3.10 |
> | Closed                       | **2.50** &pm; 0.54 | 3.17 &pm; 0.67      | 7.74 &pm; 2.82  |
> | Closed flipped               | 12.97 &pm; 1.49    | **10.49** &pm; 1.28 | 24.96 &pm; 3.82 |
>
>
> > Constraint on jPCA subspace
>
> The additional constraint we added to the existing jPCA method was to ensure a time-stable latent space during real-time calculation of the latent dynamics from the high-dimensional neural data. This is to prevent unstable and discontinuous data from appearing in the latent space, which would make it challenging to then model using a streaming dynamical systems model (whether KF, VJF, or BW). Without this constraint, a new latent representation calculated at each timepoint might be a flipped or rotated version compared to prior timepoints.
>
>
> > Comparative performance
>
> We have modified the text around Figure 1 to clarify; Figure 1 is not designed to show comparative performance between three latent space representations, merely that these representations are different, but each are learnable within a minute or so of data. This makes them all suitable for real-time dimension reduction of neural data as required for the core contribution of our work, the stimulation response mapping and optimization of new stimulations from the high-dimensional neural space to the low-dimensional latent space.

---

> > ### Author Response · Authors · 2025-11-22
> >
> > > Toy model with non-rotational dynamics
> >
> > This is an excellent suggestion. We have implemented a new toy model with linear dynamics in the x-y plane, where the ground truth response map is similar to the one in Figure 2 with stimulations displaced along the z direction. We projected this 3-dimensional data into a higher 8-dimensional space, and then used sjPCA for dimension reduction. We observed that the data thus projected into the top two dimensions of sjPCA space did not appear overly rotational, but mimicked the structure of the original 3-dimensional data. The visualization and results for this are now included in Appendix D.
> >
> >
> > **References**
> >
> > Aoi, M. C., Mante, V., & Pillow, J. W. (2020). Prefrontal cortex exhibits multidimensional dynamic encoding during decision-making. Nature neuroscience, 23(11), 1410-1420.
> >
> > Awasthi, P., Lin, T. H., Bae, J., Miller, L. E., & Danziger, Z. C. (2022). Validation of a non-invasive, real-time, human-in-the-loop model of intracortical brain-computer interfaces. Journal of Neural Engineering, 19(5), 056038.
> >
> > Batabyal, T., Brincat, S. L., Donoghue, J. A., Lundqvist, M., Mahnke, M. K., & Miller, E. K. (2024). Stability from subspace rotations and traveling waves. bioRxiv, 2024-02.
> >
> > Kuzmina, E., Kriukov, D., & Lebedev, M. (2024). Neuronal travelling waves explain rotational dynamics in experimental datasets and modelling. Scientific Reports, 14(1), 3566.
> >
> > Lebedev, M. A., Ossadtchi, A., Mill, N. A., Urpí, N. A., Cervera, M. R., & Nicolelis, M. A. (2019). Analysis of neuronal ensemble activity reveals the pitfalls and shortcomings of rotation dynamics. Scientific Reports, 9(1), 18978.
> >
> > Perich, M. G., Narain, D., & Gallego, J. A. (2025). A neural manifold view of the brain. Nature Neuroscience, 1-16.
> >
> > Rouse, A. G., & Schieber, M. H. (2018). Condition-dependent neural dimensions progressively shift during reach to grasp. Cell reports, 25(11), 3158-3168.
> >
> > Smith, J., Linderman, S., & Sussillo, D. (2021). Reverse engineering recurrent neural networks with Jacobian switching linear dynamical systems. Advances in Neural Information Processing Systems, 34, 16700-16713.
> >
> > Zhang, X., Song, Z., Shen, X., Chen, S., Huang, Y., Príncipe, J. C., & Wang, Y. (2025). Aligning Neural Population Patterns Facilitates Motor Learning Transfer. bioRxiv, 2025-05.

---

### Official Review · Reviewer_ybhx · 2025-11-01

**Soundness:** 3
**Presentation:** 2
**Contribution:** 3
**Rating:** 4
**Confidence:** 2

**Summary:**

This paper presents a real-time framework for adaptive stimulation and response modeling of latent neural dynamics. It introduces a streaming method for constructing latent neural manifolds, a nonparametric kernel-based model for stimulus-response mapping, and an optimization approach to design high-dimensional stimuli that drive low-dimensional neural dynamics in desired directions. The method is evaluated on both simulated and real neural datasets and achieves fast runtime (<100 ms), making it suitable for future closed-loop experiments.

**Strengths:**

* The integration of streaming latent space estimation, stimulus-response modeling, and stimulus optimization is innovative.
* The framework is comprehensive, covering multiple latent representations, dynamical models, and real-world constraints (e.g., non-negativity, sparsity).
* The method is computationally efficient and demonstrated to run in real time, which is critical for in vivo applications.
* Experiments and analyses are comprehensive, with both synthetic data and real-world neural data. The results are promising.

**Weaknesses:**

* The presentation of the paper is not clear.
* The impact of stimulations on behavior is not modeled or discussed, limiting the interpretability of stimulation effects in behavioral contexts.

**Questions:**

* Is there any comparison regarding the time complexity of the three methods?

---

> ### Author Response · Authors · 2025-11-22
>
> We thank the reviewer for their feedback and recognition of the contributions and strengths of our work. We address comments and questions below.
>
> > Unclear presentation
>
> Based on feedback across all reviewers, we have revised our text to improve the clarity. We would welcome any additional feedback on what might specifically still be less clear.
>
> In particular, we have added a step-by-step procedure guide at the suggestion of reviewer 739W in the main text, and below, to clarify the overall method and highlight our core contributions.
>
> **Algorithm: Real-time Adaptive Stimulation Framework**
>
> **Given:** Neural data stream {yₜ}, latent space mapping 𝒬, dynamical model f, stimulus-response model Ŝ, response delay d
>
> **Returns:** Optimized stimulus u* at decision timepoints
>
> **Initialize:** Set t ← 0, Ŝ ← ∅, stimulus history ℋ ← ∅
>
> ---
>
> **for** t = 1, 2, ... **do**
>
> 1. Observe new neural data yₜ ∈ ℝᴺ
>
> 2. Update latent projections: xₜ ← 𝒬.update(yₜ)    &nbsp;&nbsp;&nbsp;&nbsp;&nbsp; > *Observe and project to latent space*
>
> 3. x̂ₜ₊₁ ← f(xₜ)     &nbsp;&nbsp;&nbsp;&nbsp;&nbsp;> *Predict next latent state*
>
> 4. **if** stimulation delivered at time t−d (i.e., (t−d, uₜ₋ₐ) ∈ ℋ) **then**
>    - sₒᵦₛ ← xₜ − x̂ₜ     &nbsp;&nbsp;&nbsp;&nbsp;&nbsp;> *Compute observed response*
>    - Ŝ ← Ŝ.add(xₜ₋ₐ, uₜ₋ₐ, sₒᵦₛ, t)     &nbsp;&nbsp;&nbsp;&nbsp;&nbsp;> *Update kernel regression*
>
> 5. **else**
>    - f ← f.update(xₜ, xₜ₋₁)     &nbsp;&nbsp;&nbsp;&nbsp;&nbsp;> *Update dynamics model*
>
> 6. **end if**
>
> 7. **if** new stimulation desired at time t **then**
>    - **Given:** target direction v ∈ ℝᵏ
>    - Define loss function:
>
>      𝓛(u) = −(vᵀs(u))/(‖v‖‖s(u)‖) + λ₁(‖u‖₀ᵐᵃˣ − ‖u‖₁)     &nbsp;&nbsp;&nbsp;&nbsp;&nbsp;> *Optimization problem in (8)*
>
>      where s(u) = Ŝ(xₜ, u, t)     &nbsp;&nbsp;&nbsp;&nbsp;&nbsp;> *Predicted response via learned mapping*
>
>    - u* ← argmin_{u ∈ [0,1]ᴺ} 𝓛(u)     &nbsp;&nbsp;&nbsp;&nbsp;&nbsp;> *Solve with box constraints*
>    - Deliver stimulation u* to neural system
>    - Add (t, u*) to ℋ     &nbsp;&nbsp;&nbsp;&nbsp;&nbsp;> *Track pending stimulation*
>
> 8. **end if**
>
> **end for**
>
>
> > Impact of stimulations on behavior
>
> This is an excellent suggestion by the reviewer to include modeled effects of behavioral responses, in addition to the already presented latent neural responses. We should note that there is existing literature on optimization of neural or peripheral stimulations for behavior, ignoring the effects on the neural activity or latent space (Brocker, 2017; Yu, 2020; Bonizzato, 2023). It is typically possible to treat the mapping between neural stimulation and behavior as a black box and apply e.g. Bayesian optimization to the 1-dimensional problem of e.g. maximizing ankle flexion.
>
> What is particularly interesting in our framework is that the stimulus response map we learn (for the latent neural space) has, either by design or empirically in real stimulation data, correlations across the space that make the map learnable via our regression method. This may not in fact be the same case for behavior, especially as there is not always a one-to-one mapping from neural latent space to behavior. While some simple low-dimensional behaviors, as in motor reaching tasks, might have this spatial structure, other behaviors, especially more complex ones as in free-moving or cognitive outcome tasks, might not.
>
> Based on the suggestion of reviewer JwEk, we have also added new experiments using real stimulation datasets. In one of these (Draelos, 2025) there is also behavior included along with stimulated neural activity. We took this behavior (tail movement) and incorporated it into our regression method to learn a mapping from stimulations at locations in the neural latent space to behavior responses. We have added the results of this to Appendix C where we plot the error in prediction for each stimulation in the neural latent space.

---

> > ### Author Response · Authors · 2025-11-22
> >
> > > Time complexity
> >
> > We thank the reviewer for this question and have expanded Appendix H detailing the time complexity of each component of our framework. In practice, even with longer experiments (20,000 timepoints), we saw only small increases in end-to-end runtime, with averages below 100ms per step.
> >
> > For the stimulation response model, we use a form of kernel regression that minimizes loss in polynomial time (Cesa-Bianchi, 2015) since we learn a set of coefficients fairly efficiently. For the online or streaming setting, complexity via gradient descent is O(dN) where d is the dimensionality of the space (Li, 2023). Here, since d is small and N itself grows relatively slowly (at the rate of stimulation events), this is easily managed.
> >
> > For the stimulation design algorithm, we implemented the constrained optimization problem with the L-BFGS-B method (which is the limited memory and box-constrained variant of BFGS; Zhu, 1997) which has complexity O(iN) where i is the number of iterations and N is the dimensionality of the high-dimensional stimulation vector u (typically, the number of neurons). This will not grow as a function of time throughout an experiment, and so should be constant for a given neural population.
> >
> >
> >
> > **References**
> >
> > Bonizzato, M., Hottin, R. G., Côté, S. L., Massai, E., Choinière, L., Macar, U., ... & Dancause, N. (2023). Autonomous optimization of neuroprosthetic stimulation parameters that drive the motor cortex and spinal cord outputs in rats and monkeys. Cell Reports Medicine, 4(4).
> >
> > Brocker, D. T., Swan, B. D., So, R. Q., Turner, D. A., Gross, R. E., & Grill, W. M. (2017). Optimized temporal pattern of brain stimulation designed by computational evolution. Science translational medicine, 9(371), eaah3532.
> >
> > Cesa-Bianchi, N., Mansour, Y., & Shamir, O. (2015, June). On the complexity of learning with kernels. In Conference on Learning Theory (pp. 297-325). PMLR.
> >
> > Li, J., & Liao, S. (2023, July). Nearly optimal algorithms with sublinear computational complexity for online kernel regression. In International Conference on Machine Learning (pp. 19743-19766). PMLR.
> >
> > Yu, C., Cassar, I. R., Sambangi, J., & Grill, W. M. (2020). Frequency-specific optogenetic deep brain stimulation of subthalamic nucleus improves parkinsonian motor behaviors. Journal of Neuroscience, 40(22), 4323-4334.
> >
> > Zhu, C., Byrd, R. H., Lu, P., & Nocedal, J. (1997). Algorithm 778: L-BFGS-B: Fortran subroutines for large-scale bound-constrained optimization. ACM Transactions on mathematical software (TOMS), 23(4), 550-560.

---

### Official Review · Reviewer_JwEk · 2025-11-03

**Soundness:** 1
**Presentation:** 3
**Contribution:** 2
**Rating:** 2
**Confidence:** 4

**Summary:**

This paper develops a real-time stimulation framework targeting alterations in neural population / latent state space. The framework requires 1) an online system identification component, which the authors propose a novel, realtime variant of jPCA to test along online versions of SVD and ICA, 2) a state estimation / prediction component, such as Kalman Filter, 3) a learnable stimulation-response model, which is taken to be kernel regression, and 4) a optimization routine to find the best stimulation pattern given a goal perturbation direction in state space. The proposed method was tested on a synthetic rotational system, and two experiments in which simulated perturbations were added to offline recorded neural data.

**Strengths:**

- the problem of realtime perturbation in neural state space is clearly motivated, important, and systematically described
- the proposed framework and its various modules is comprehensive and intuitively appealing
- the paper is clearly and concisely written overall

**Weaknesses:**

- the modeled perturbation in both the synthetic system and real data as a independent increase and stationary decay seem too simplistic, and it was a bit confusing in the different experiments whether the simulated stimulation effect is applied to the data space or the latent space.
- “real data” experiments also only consisted of simulated perturbations with simple decaying responses added onto real data, not actual perturbation experiments, which is not mentioned until 5 and a half pages in. Critically, it’s unclear how the natural evolution of neural populations in the data interacts with the intended / feasible directions in the stimulation experiments (e.g. lines 410-423).
- the key contributions were diluted with the different novel components introduced. For example, it was unclear whether the streaming jPCA method / rotational latent model was necessary or performed better than the other two models, or if in the experiments the 3 models were always used, and if so, when the detected transitions were.
- overall, I believe the current experimental design, in particular with the simulated perturbations on real data, does not demonstrate the validity nor value of the proposed framework, nor is it clear the key contributions of the various components and their respective importance for the whole framework.

**Questions:**

- how stable is the online centering, i.e., mean/std estimation, for the various modalities of neural data?
- Fig1c shows that the space most likely to give the best predictive probability (bright colors) coincide with low-magnitude parts of the flow field in 1b. Essentially, the parallel estimates of each system online gives the most likely system as the one with the least movement, which doesn’t seem ideal? Am I misinterpreting something here?
- what happens if in the synthetic system the stimulation acts on multiple coordinates instead of just x3 (eq. 9)?
- in section 4.2 / figure 4, what did the algorithm-designed stimulations look like? Given that the first PC (Q0) was the target, stimulating random individual / groups of neurons seems to be an inappropriate hypothesis. Intuitively, the naive hypothesis would be that stimulation patterns closely resembling the loading vectors would be maximally effective. Was this the case or tested?
- line 417: it was found that some random directions were easy whereas others were not, did the “easy” directions coincide with the real data evolution at the time? In other words, was it easier to induce a movement in the direction that the data was naturally moving towards in the first place?

---

> ### Author Response · Authors · 2025-11-22
>
> We thank the reviewer for their comprehensive review and good summary of our work and its strengths. We address comments and questions below.
>
> >Modeled perturbation
>
> Our stimulation framework is designed to learn high-dimensional stimulation vectors that produce a desired effect in the latent space. **All** our simulated stimulations are meant to follow that paradigm, so we are applying the stimulations directly to the high-dimensional neural data (data space). We then apply dimension reduction to observe the resultant effects in the latent space. We have clarified this to reduce confusion for each experiment in the main text, and have added an overview of the complete procedure in Algorithm 1.
>
> We selected an autoregressive form of modeling responses to stimulations (in the data space) to mimic real perturbations used in experiments (Kim, 2017; Inagaki, 2019; Daie, 2021; Zhang, 2023; Chen, 2025; Triplett, 2025). Similar forms have also been used in other models of the effects of stimulations (Wagenmaker, 2024). The data we showed in Figure 3 is from neural data recorded using calcium fluorescence imaging and part of the temporal profile is determined by the calcium indicators used. Electrophysiological responses would look different (an increase in the number of spiking events); though if binned into firing rates, as is commonly done, this would again look similar to the calcium traces in response to stimulations. As detailed below, we also now include results from real stimulation experiments and a visualization of similar calcium responses that align with our simulated ones.
>
>
> >Real stimulation experiments
>
> While we don’t have the required experimental capabilities for conducting stimulations in vivo ourselves, we wanted to provide an algorithmic and methodological contribution for designing stimulations for current and future neuroscience experiments. There is an exciting opportunity for using stimulations to learn better models of neural representations via these causal interventions; yet most stimulations are simple in nature due to the difficulty of designing them live and adapting to their effects within interconnected neuronal circuits. Our goal is to provide this framework for experimentally feasible stimulations learned in real time, and demonstrate its utility in multiple scenarios for generalization to future experiments.
>
> Since submission, we have identified 2 datasets that conducted direct neuronal stimulations and provide neural time traces and stimulation events that we used for additional validation of our stimulus-response regression model. While no dataset exists for stimulations targeted at neural latents, motivating our core contributions, we find it helpful for demonstrating our method on neural data that had actually been stimulated, and not in simulation.
>
> The first dataset contains two-photon calcium fluorescence traces during photostimulation of motor cortex neurons in mice during a memory guided response task (Daie, 2021). The experiment consisted of photostimulations with three different patterns of small groups of neurons (fewer than 10). As the data are discontinuous trials of single stimulation events, we randomly interleaved them to create a continuous data stream and then proceeded to learn a stimulus-response map with our method. Our regression model obtained an average prediction error of 1.84 compared to 2.63 in the blind comparison case (lower is better; similar to Figure 3c). We have added this new result to the manuscript currently in Appendix C but are working to expand current Figure 3 to include this result as well.
>
> The second dataset contains two-photon calcium fluorescence traces during photostimulation of head-fixed larval zebrafish (Draelos, 2025).  The experiment consisted of photostimulations with different targets and was continuous across time. Our regression model obtained an average prediction error of 4.63 compared to 5.67 in the comparison case. We have added this result, similar to Figure 3c, to Appendix C, along with a visualization of the stimulated neural traces in the original high-dimensional space (as in Fig. 3a) and in the latent space (as in Fig. 3b), and will include a revised version of Figure 3 incorporating these results from real stimulations experiments as well.

---

> > ### Author Response · Authors · 2025-11-22
> >
> > > Neural dynamics interaction with intended stimulation direction
> >
> > This is a very interesting question that we did not explicitly consider. We applied stimulations at random times, and consequently random locations in the neural latent space, to provide the most fair and well-covered set of possible stimulation responses for our model to learn. Infeasible directions are dictated by constraints in the high-dimensional space (stimulations don’t cause negative neural activity; at least not with most experimental techniques), mapped back to the latent space. These would be consistent regardless of the evolution of neural dynamics or location in the latent space at which the stimulation is delivered.
> >
> > Based on the reviewer’s feedback, we have added an analysis of the interaction between ongoing neural dynamics (direction of non-stimulated neural activity in the latent space) and the intended stimulation vector (e.g., along the first principal component, or any feasible direction). We used the (O’Doherty, 2024) dataset from main Figure 4, generated stimulations at different timepoints, and calculated the angle between the current direction of motion in the latent space and the desired perturbation (‘Dynamics angle’), and the angle between the actually observed response and the desired perturbation (‘Response angle’). We observed mostly what we expected; that for a wide variety of possible dynamics angles, if the stimulation was feasible, the response angle was small (well aligned with the target) regardless of ongoing motion. We did not conclude that there was a statistically significant correlation between the angle between ongoing dynamics and the desired target, and the angle between the observed response and the desired target (Feasible: p=0.79, Q_0: p=0.83). This analysis is now in Appendix G.
> >
> > Finally, we also include a visualization in Appendix C of the error in predictions mapped onto different locations in the latent space (first two dimensions of proSVD representation) for the two real stimulation datasets we now include (Daie, 2021; Draelos, 2025). We found it interesting to see if some locations were easier to learn responses from than others. There was a weak spatial grouping, but future work might follow up on this type of analysis to learn more about how neural dynamics and traversals in the latent space may interact with stimulation directions.

---

> > > ### Author Response · Authors · 2025-11-22
> > >
> > > > Differences between latent spaces
> > >
> > > This is a very important point to clarify; we thank the reviewer for their specific example and have updated the text to resolve confusion and added a new results figure for more insight.
> > >
> > > While we view our core contribution as the closed-loop design of stimulations in the high-dimensional space to produce effects in the latent space, we wanted our framework to be usable in a general way. We thus chose to implement the necessary preprocessing steps of (1) dimension reduction to the latent space and (2) modeling the dynamics of the non-stimulated latent neural activity. Here we mostly leveraged existing methods to show how our method could be integrated with each of them, but also to show that our method is robust to the specific neural representation chosen for an individual experiment.
> > >
> > > Because our method is designed for closed-loop and real-time experiments, we also required that preprocessing steps be executable also in real time. We selected proSVD and mmICA as streaming methods that represented the neural data under different assumptions: high-variance directions, and independent components. As jPCA is a common neural representation framework, we wanted to add it as well, and so developed sjPCA as a streaming-capable and time-stabilized implementation. While the interpretation of the rotational dynamics learned by jPCA is debatable (e.g, Lebedev, 2019; Kuzmina, 2024), jPCA is widely used in neuroscience (see reviews in Kuzmina, 2023; Perich, 2025), even beyond motor contexts (e.g., prefrontal cortex in Aoi, 2020).
> > >
> > > We did not compare each set of latent space and dynamical model combination, nor explore this in great detail in the original text. Here, at the reviewer’s suggestion, we have now expanded Appendix E and provided results for the experiments run in Figures 3 and 4. Across all datasets and experiments (open and closed-loop), we found that the proSVD latent space yielded the best results on average, independent of dynamical model. However, there were certain conditions we tested (open-loop, flipped response map) where sjPCA provided the best latent space representation. We also found that the Kalman filter provided the best results in the majority of the cases studied here. Below we present one example set of results on the (O’Doherty, 2024) dataset used in Figure 4 where we design stimulations using open-loop, open-loop with a flipped mapping, closed-loop, and a closed-loop with flipped mapping between stimulation vectors and responses. A complete listing of results is in Appendix E.
> > >
> > > ||s_{obs} – S_{i-1}(xi, ui, ti)||
> > >
> > > | Stim. Cond. / Lat. Sp. Model | sjPCA              | proSVD              | mmICA           |
> > > |------------------------------|--------------------|---------------------|-----------------|
> > > | Open                         | 3.04 &pm; 0.63     | **2.25** &pm; 0.36  | 4.53 &pm; 0.58  |
> > > | Open flipped                 | 13.07 &pm; 1.57    | **10.21** &pm; 1.19 | 25.46 &pm; 3.10 |
> > > | Closed                       | **2.50** &pm; 0.54 | 3.17 &pm; 0.67      | 7.74 &pm; 2.82  |
> > > | Closed flipped               | 12.97 &pm; 1.49    | **10.49** &pm; 1.28 | 24.96 &pm; 3.82 |
> > >
> > >
> > > Because these are all streaming methods, it is straightforward to run them in parallel, or even switch among them to leverage the best predictor at any given timestep. We have added a figure to Appendix A to show how at each timepoint a different representation might be used as the best predictor. Using such an adaptive selection of spaces improved average log predictive probability from -1.72 with the best space (proSVD) to -1.01 using all three spaces. We anticipate that this parallelization and streaming capabilities would enable future experiments where best-modeled responses lie in a single latent space, and stimulations are conducted to disambiguate candidate spaces.
> > >
> > > > Stability in online centering
> > >
> > > Excellent question. We looked at how much the centers of the learned spaces changed over time for both the (O’Doherty, 2024) electrophysiological neural data and the (Draelos, 2025) calcium fluorescence neural data. We quantified both the relative change in the centers as well as the relative change in the covariances describing the proSVD space. For the first dataset, we found that the stepwise differences in the estimate of the mean are less than .5% of the magnitude of the total mean for all timepoints after ~16s, and the frobenius norms of the stepwise differences in the estimate of the covariance matrix of the full-dimensional data are less then 1% of the final covariance estimate for all times after 88s. For the second dataset, we similarly found that differences in the center were down to 0.5% after 113s, and covariances down to 1% after 690s. Overall we would consider these quite stable. Full plots for these analyses are in Appendix A.

---

> > > > ### Author Response · Authors · 2025-11-22
> > > >
> > > > > Space with best predictive probability
> > > >
> > > > This is a great question raised here that we did not initially characterize. In the center, the flow appears consistently small. This is average flow, which means it can contain multiple large but opposing flows that are canceled out in the visualization we used. We cross-checked by adding a data density plot to determine if this was just a low data area and found that it is not. In addition to the above analysis showing the benefit of switching between these spaces, we also now include the data density plots as a comparison to Figure 1 in Appendix A.
> > > >
> > > >
> > > > > Stimulation on multiple coordinates
> > > >
> > > > This is a great question that could be explored with our toy model. (We did employ multi-dimensional stimuli in all the simulated stimulation experiments.) We have added a new experiment in the toy model where stimulations are applied to all 3 dimensions, and demonstrate that our response model is still able to predict the results of those stimulations, very similar to the existing results in Figure 2. Our regression method obtained an average error of 1.01 pm 0.30 compared to 1.33 pm 0.42 in the blind comparison method during the last rotational block (as in the last portion of Figure 2e). The full results for this toy model experiment are now presented in Appendix D.
> > > >
> > > >
> > > > > Algorithm-designed stimulations
> > > >
> > > > The reviewer is correct; a good hypothesis would be that stimulation vectors similar to the loading vectors would produce movement along the first principal component. The first difficulty with delivering such stimuli lies in the constraints: we typically can’t apply a negative stimulation, and can only stimulate up to a certain number of neurons at once. The first principal component might be instead dominantly composed of negative and positive loadings across too many neurons. So our optimization framework would find the closest feasible stimulation to apply instead.
> > > >
> > > > The second difficulty is if stimulations don’t evoke an identity effect but instead are governed by a more complex response function dependent on network interactions and current dynamical state. We model this via our closed-loop framework, where we do not assume that a stimulation vector, that might be exactly along the first PC after dimension-reduction, indeed results in a neural latent response in that same direction. This can be the case in real neuroscience experiments where repeated stimulations can produce diverse (inconsistent) responses.
> > > >
> > > > We included random stimulations as a comparison to highlight the type of responses we might expect to see for such stimulations, as well as because such stimulations are often used in current experimental designs. We did not expect those stimulations to always evoke responses aligned with the first PC, though we note some rare few did produce such responses (Figure 4b). The effects of single and random sets of stimulations are generally not aligned with ongoing neural dynamics, which matches what is often observed in practice. This was important to also demonstrate in our system to validate against what is known from the literature, and further motivate why adaptive optimization from high-dimensions to low-dimensions is necessary for effective stimulations that can drive neural dynamics.
> > > >
> > > > We have included a new visualization of the stimulations designed for the (Zong, 2021) dataset case, where stimulations at one location in the latent space were simulated in all directions in the first principal plane, and resultant designed stimulations are generated for each of those directions. Some directions are not feasible (likely due to requiring inhibition), visualized by no designable stimulation vectors, whereas other directions are very feasible. This analysis is now included in Appendix G.
> > > >
> > > >
> > > > > Easy vs hard directions
> > > >
> > > > We discussed this in more detail in our response above on ‘Neural dynamics interaction with intended stimulation direction’; but generally speaking, no. The dominating factor for ‘easiness’ or feasibility is determined by the desired direction of movement in the latent space (the vector v). This makes some intuitive sense, as some directions are most similar to the e.g. loading vectors (with variations due to sparsity constraints).
> > > >
> > > > We did note some interesting counter examples in the real datasets we examined where although most repeated stimulations (at different locations in the latent space) produced robustly similar responses, occasional stimulations at the extent of the latent space produced little to no response. We find it likely that this can be attributed to refractory periods (neurons are already maximally active and stimulating them isn’t effective) and other non-modeled network effects. What our method provides is the ability to learn such differences in responses at different locations in the latent space, and use that information on the fly to either avoid stimulating there or design different stimulations, if feasible ones exist.

---

> > > > > ### Author Response · Authors · 2025-11-22
> > > > >
> > > > > **References**
> > > > >
> > > > > Aoi, M. C., Mante, V., & Pillow, J. W. (2020). Prefrontal cortex exhibits multidimensional dynamic encoding during decision-making. Nature neuroscience, 23(11), 1410-1420.
> > > > >
> > > > > Chen, I. W., Chan, C. Y., Navarro, P., de Sars, V., Ronzitti, E., Oweiss, K., ... & Emiliani, V. (2025). High-throughput synaptic connectivity mapping using in vivo two-photon holographic optogenetics and compressive sensing. Nature Neuroscience, 1-13.
> > > > >
> > > > > Daie, K., Svoboda, K., Druckmann, S. (2021). Dataset supporting "Targeted photostimulation uncovers circuit motifs supporting short-term memory". Janelia Research Campus. Dataset located at: https://doi.org/10.25378/janelia.13546157.v1
> > > > >
> > > > > Draelos, A., Loring, M. D., Nikitchenko, M., Sriworarat, C., Gupta, P., Sprague, D. Y., ... & Naumann, E. A. (2025). A software platform for real-time and adaptive neuroscience experiments. Nature Communications, 16(1), 9909. Dataset located at: https://dandiarchive.org/dandiset/001569
> > > > >
> > > > > Inagaki, H. K., Fontolan, L., Romani, S., & Svoboda, K. (2019). Discrete attractor dynamics underlies persistent activity in the frontal cortex. Nature, 566(7743), 212-217.
> > > > >
> > > > > Kim, S. S., Rouault, H., Druckmann, S., & Jayaraman, V. (2017). Ring attractor dynamics in the Drosophila central brain. Science, 356(6340), 849-853
> > > > >
> > > > > Kuzmina, E., Kriukov, D., & Lebedev, M. (2024). Neuronal travelling waves explain rotational dynamics in experimental datasets and modelling. Scientific Reports, 14(1), 3566.
> > > > >
> > > > > Lebedev, M. A., Ossadtchi, A., Mill, N. A., Urpí, N. A., Cervera, M. R., & Nicolelis, M. A. (2019). Analysis of neuronal ensemble activity reveals the pitfalls and shortcomings of rotation dynamics. Scientific Reports, 9(1), 18978.
> > > > >
> > > > > O’Shea, D. J., Duncker, L., Goo, W., Sun, X., Vyas, S., Trautmann, E. M., ... & Shenoy, K. V. (2022). Direct neural perturbations reveal a dynamical mechanism for robust computation. bioRxiv, 2022-12.
> > > > >
> > > > > Triplett, M. A., Gajowa, M., Antin, B., Sadahiro, M., Adesnik, H., & Paninski, L. (2025). Rapid learning of neural circuitry from holographic ensemble stimulation enabled by model-based compressed sensing. Nature Neuroscience, 1-12.
> > > > >
> > > > > Perich, M. G., Narain, D., & Gallego, J. A. (2025). A neural manifold view of the brain. Nature Neuroscience, 1-16.
> > > > >
> > > > > Wagenmaker, A., Mi, L., Rozsa, M., Bull, M., Svoboda, K., Daie, K., ... & Jamieson, K. G. (2024). Active learning of neural population dynamics using two-photon holographic optogenetics. Advances in Neural Information Processing Systems, 37, 31659-31687.
> > > > >
> > > > > Zhang, J., Hughes, R. N., Kim, N., Fallon, I. P., Bakhurin, K., Kim, J., ... & Yin, H. H. (2023). A one-photon endoscope for simultaneous patterned optogenetic stimulation and calcium imaging in freely behaving mice. Nature Biomedical Engineering, 7(4), 499-510.

---

### Author Response · Authors · 2025-11-22

We thank all reviewers for their time and effort in reviewing our manuscript. We found many of the questions, suggestions, and ideas raised in the reviews to be useful not just for clarifying our existing work but also for pushing us to consider new implications or use cases for our methods. We think all of this has substantially strengthened our manuscript.

Based on the feedback, we have improved our work with a few new additions:
* A step-by-step procedure guide in Algorithm 1
* New experiments on data from real stimulation experiments showing the effectiveness of our stimulation response model across toy data, real data with simulated stimulations, and real data with stimulation events.
* New experimental results detailing the effect of choice of latent space and dynamical model on the regression model performance.
* New analyses of our stimulation optimization results including relaxation of constraints, interactions between ongoing dynamics and target vector, and relationship between feasibility and designed stimulations.
* New experiments on real stimulation data modeling the predicted effects of stimulations on behavior, in addition to the effects on latent neural dynamics.

---

### Meta-Review · Area_Chair_g2kf · 2026-01-05

**Summary:**

The goals of this paper are to develop real-time methods for tracking and guiding neural activity through neural stimulation and stimulus presentation. The method consists first of a dimensionality reduction and latent state estimation step, updating the model of a latent dynamical system, and then updating the estimate of the response to a stimulus/stimulation. With the model fit, the simulation response was optimized over a large parameter sweep. The authors apply their method to a toy model, as well as two real datasets representing well-used recording technologies: electropysiology and calcium imaging data.

The main points that motivate the decision are the combination of lack of clarity with the limited realistic real data examples. Another important point that was better addressed, but could be better incorporated into the manuscript was the reliance on jPCA. I agree with the authors that some people use it, but the emphasis on rotations is an assumption that might not always hold. The additional appendix does help with this point which is why it is a lesser point than the other two.

**Reviewer Concerns:**

The main reviewer concerns were:
 1) Lack of real perturbations in the real neural recording data
 2) Simplistic model of neural perturbations
 3) Lack of discussion on behavioral responses (real or expected)
 4) Lack of clarity, including on model contributions and which were the main novel aspects
 5) Over-reliance on jPCA

The authors provided a number of clarifications, and I agree with a numner of their responses. I do also agree with the reviewers that the simple perturbation model does need stronger justification to warrant all the methods development, which is unfortunately not available with the given data. Thus I consider points 1/2 still outstanding. Moreover after reading the later version of the manuscript I feel that the contributions are still not fully clear and additional revision might help lead the reader to better see what the authors are intending to disseminate.

**Reviewer Scores:**

The reviewers initially gave a 2,4,2,6. Reviewer 739W was the only one to respond and maintained a score of 6. I thus estimate that the remainder would not sufficiently raise their scores given the response.

---

### Decision · Program_Chairs · 2026-01-26

Reject